# Near surface structure of Sodankylä area in Finland, obtained by passive seismic interferometry

Nikita Afonin[1,5], Elena Kozlovskaya[1,2,4], Suvi Heinonen[2], Stefan Buske[3]

[1]Oulu Mining School, POB-3000, FIN-90014, University of Oulu, Finland
[2]Geological Survey of Finland, P.O. Box 96, FI-02151, Espoo, Finland
[3]TU Bergakademie Freiberg, Institute of Geophysics and Geoinformatics, Freiberg, Germany
[4]Sodankylä Geophysical Observatory, University of Oulu, Finland
[5]N. Laverov Federal Center for Integrated Arctic Research of the Ural Branch of the Russian Academy of Sciences, Arkhangelsk, Russia

*Correspondence to*: Nikita Afonin (nikita.afonin@oulu.fi)

**Abstract.** Controlled-source seismic exploration surveys are not always possible in nature-protected areas. As an alternative, application of passive seismic techniques in such areas can be proposed. In our study, we show results of passive seismic interferometry application for mapping the uppermost crust in the area of active mineral exploration in Northern Finland. We
are utilizing continuous seismic data acquired by Sercel Unite Wireless multichannel recording system along several profiles during XSoDEx (eXperiment of SOdankylä Deep Exploration) multidisciplinary geophysical project. The objective of XSoDEx was to obtain a structural image of the upper crust in the Sodankylä area of Northern Finland in order to achieve a better understanding of the mineral system at depth. The key experiment of the project was a high-resolution seismic reflection experiment. In addition, continuous passive seismic data was acquired in parallel with reflection seismic data acquisition. Due
to this, the length of passive data suitable for noise cross-correlation was limited from several hours to couple of days. Analysis of the passive data demonstrated that dominating sources of ambient noise are non-stationary and have different origin across the XSoDEx study area. As the long data registration period and isotropic azimuthal distribution of noise sources are two major conditions for Empirical Green's Functions (EGFs) extraction under diffuse field approximation assumption, it was not possible to apply the conventional techniques of passive seismic interferometry. To find the way to obtain EGFs, we used
numerical modelling in order to investigate properties of seismic noise originating from sources with different characteristics and propagating inside synthetic heterogeneous Earth models representing real geological conditions in the XSodEx study area. The modelling demonstrated that scattering of ballistic waves on irregular shape heterogeneities, such as massive sulphides or mafic intrusions, could produce diffused wavefield composed mainly of scattered surface waves. In our study we show that this scattered wavefield can be used to retrieve reliable Empirical Green Functions (EGFs) from short-term and non-
stationary data using special techniques. One of the possible solutions is application of "signal-to-noise ratio stacking" (SNRS). The EGFs calculated for the XSoDEx profiles were inverted, in order to obtain S-wave velocity models down to the depth of 300 meters. The obtained velocity models agree well with geological data and complement the results of reflection seismic data interpretation.

## 1 Introduction

Exploration of new mineral deposits is an important task because the modern world needs many types of minerals for functioning (Reid, 2011). Most of the shallow mineral deposits around the world nowadays are well known and exploration of new deep mineral deposits becomes more difficult (Vasara, 2018). That is why the cost-effective exploration techniques are required. Moreover, there is the problem that application of controlled-source seismic exploration is not always possible in nature-protected areas. Particularly in the Arctic areas, non-invasive, environmentally friendly exploration is relevant. As an

alternative, application of passive seismic techniques in such areas has been proposed. The main advantage of passive seismic methods is possibility to study the subsurface in remote areas with minimum impact to environment (Polychronopoulou et al., 2020).

Passive seismic interferometry is cost-efficient methodology with relatively simple setup of field experiments. This methodology allows retrieving impulse response of a medium, called Empirical Green's Function (EGF) from ambient seismic

noise recorded at two receivers, assumed that the noise field is diffuse (Lobkis and Weaver, 2001; Campillo and Paul, 2003). If this condition is satisfied, it is possible to retrieve both surface and body waves from seismic noise using either cross-correlation, autocorrelation, deconvolution or cross-coherence of seismic records (Wapenaar et al., 2011). As shown in Ricker and Claerbout (1999), the condition of diffuse noise field is satisfied if the noise sources are distributed isotropically around seismic recorders and the noise registration time is long enough. The methodology to retrieve EGFs, using cross-correlation

or autocorrelation of ambient seismic noise, has been successfully applied in numerous studies (e.g. Shapiro and Campillo, 2004; Roux et al., 2005; Ruigrok et al., 2011; Draganov et al., 2009; Poli et al., 2012; Tibuleac et al,. 2012; Wang et al., 2015; Taylor et al., 2016; Afonin et al., 2017; Oren and Nowack, 2016; Romero et al., 2018). In addition to ambient seismic noise interferometry, the coda wave interferometry was proposed (Aki and Richards, 2002;Campillo et al., 2003; Snieder et al., 2002; Snieder, 2006). This methodology is also based on diffuse filed approximation and it is widely used for different

purposes, such as estimating nonlinear behaviour in seismic velocity (Snieder et al., 2002), monitoring of stress changes inside the studied medium (Grêt et al., 2005, 2006) and determination of the third order elastic constants in a complex solid (Payan et al., 2009). Numerous studies describe results of successful application of passive seismic interferometry for exploration and other applied geophysical purposes (e.g. Cheraghi et al., 2017; Roots et al., 2017; Dantas et al., 2018; Abraham et al., 2019; Polychronopoulou et al., 2020; Planès et al., 2020). In spite of the numerous studies, there is the problem that the conditions

of the diffuse noise field from sources outside an observation area are difficult to satisfy in many practical situations. One important condition is isotropic (that is, when the waves arrive from all azimuths) and homogeneous (when the waves arrive from all azimuths and have near the same energy) azimuthal distribution of noise sources. In order to satisfy this condition, long registration time is required. However, the diffuse wavefield for EGFs evaluation can exist also under other conditions. Wapenaar (2004) demonstrated that EGFs could be retrieved from cross-correlation of two recordings of a wavefield at

different receiver locations at the free surface in the case when diffuse wavefield is produced by many uncorrelated sources

inside the subsurface. Wapenaar and Thorbecke (2013) also considered conditions for EGFs retrieval from ambient noise originating from a directional scatterer in a homogeneous embedding medium that is illuminated by a directional noise field. In our paper we examine application of passive seismic interferometry for the case when noise is strongly directional and receiver array is semi-linear. For this, we use continuous seismic data recorded during XSoDEx (eXperiment of SOdankylä Deep Exploration) project in northern Finland (Buske et al., 2019). We analyse the ambient seismic noise recorded by stations located in different sub-regions of the XSoDEx study area, in order to understand spatial and temporal distribution of noise sources. We perform numerical modelling of propagation of seismic wavefield corresponding to identified noise sources through synthetic models that represent certain types of heterogeneities in our study area (mafic intrusions, faults, massive sulfides). We show by numerical modelling that direct waves, generated by various sources are scattered on these heterogeneities and produce diffused wavefield. Therefore, EGFs can be retrieved by cross-correlation of this wavefield recorded at different locations (Wapenaar, 2004; Wapenaar and Thorbecke, 2013; van Manen et al., 2005). For evaluation of EGFs, we apply the method of passive seismic interferometry with so-called signal-to-noise ratio stacking (SNRS) algorithm (Afonin et al., 2019). We show results of application of passive seismic interferometry for mapping the uppermost crust in the XSoDEx study area.

## 2 Experiment description

The XSoDEx seismic survey was conducted in the Central Lapland Greenstone belt in Northern Finland, around Sodankylä region (Figure 1). The area is famous for its mineral deposits, including operating Kittilä Gold mine west from the survey area and Kevitsa Ni-Cu mine that also has significant amounts of platinum, palladium, gold and cobalt. The seismic survey lines are crossing varying geology including outcrops of Archean basement and layered mafic intrusions (Buske et al., 2019). Within the XSoDEx project, Geological Survey of Finland, TU Bergakademie Freiberg and University of Oulu acquired seismic reflection and refraction data using the Vibroseis © truck of TU Bergakademie Freiberg and partly explosive sources during July and August 2017 resulting in four seismic profiles of total length of approximately 80 km recorded along existing roads (Figure 1): Pomokairantie line (about 37.5 km); Alaliesintie line (about 14 km); Sakatti line (about 20 km); Kuusivaarantie line (about 16 km). The seismic reflection data were recorded in a roll-along scheme by a maximum 3.6 km long spread of cabled vertical-component geophones with 10 m spacing. The seismic reflection layout was designed to map crustal structures down to a minimum of 3 km depth in detail. The seismic refraction data were recorded separately along all reflection profiles by the other set of instruments that included 60 vertical component 5 Hz geophones and 40 three-component MEMS accelerometers with Sercel Unite wireless autonomous data acquisition units by Sercel Ltd. Receiver spacing for Sercel Unite wireless system was approximately 160 meters and the whole spread for recording refraction data was intended to be about 12 km long. The sampling rate was typically 250 sps, except for one high-resolution 1 km long profile. For this profile, the distance between recording units was 10 m and the sampling rate was 500 sps. The plan was to keep 75 RAUs (Remote Acquisition Units) recording continuous data at the spread while charging the remaining 25 RAUs at the accommodation of the field team. When the Vibroseis © truck moved to the large enough distance from the 25 receivers at the beginning of the

spread, those RAUs were demobilised to be charged during the nighttime. Correspondingly, the charged 25 RAUs were
deployed at the end of the line. In the field conditions, the deployment and demobilisation time schedule was adopted to take
into account the progress of active seismic experiment. Refraction seismic data was extracted from the continuous data during
periods when active seismic source was in operation, while the passive seismic data were accumulated during periods when
active seismic source was not working (nighttime, spare days). Generally, the length of time intervals for continuous passive
data recording was about 8-9 hours. Thus, the XSoDEx experiment provided a good opportunity to verify results of passive
seismic interferometry with controlled-source seismic data, to identify limitations of this technique in areas of generally low
level of high-frequency anthropogenic noise and to propose possible improvements of known techniques.

## 3 Ambient seismic noise in XSoDEx study area

As the population density in northern Finland is low, this results in low level of anthropogenic high-frequency noise in the
XSoDEx study area. In addition to microseismic noise, there are several local industrial noise sources: Kevitsa Mine, traffic
noise from the roads and noise from waterpower plants of Kitinen River.

For estimation and comparison of noise level for different XSoDEx profiles, we used vertical components of ambient seismic
noise recorded by wireless seismic receivers with 5 Hz geophones. The data was not acquired during night-time on Sundays,
when the anthropogenic activity is minimal. The noise spectra were estimated using the whole periods of continuous data
(usually about 8-9 hours), in order to find averaged characteristics of ambient noise. The sources of anthropogenic noise in our
study area may be considered in an approximation of quasi-stationary noise sources during considered time intervals. For
example, mining activity in Kevitsa mine is not changing during long time and mining machinery, excavation, transportation
of ore, blasting in the production area are producing seismic signals of similar amplitudes and frequencies for different time
periods. The dams can be also considered as sources of quasi-stationary signals. Generally, the roads may be used by transport
of different types and, as result, the traffic noise may have some temporal differences. Nevertheless, the roads in our study
area are characterized by generally low traffic. For example, the traffic along all the roads, along which the XSoDEx seismic
data was acquired, was several cars per day. The highway no. 4 (Fig. 1) is characterised by slightly larger traffic during
weekdays, but it is still very quiet during night- time and weekends.

As one can see in Figure 2, the noise amplitude and its frequency spectra differ significantly at stations located at different
sites in XSoDEx area. Station V1 was installed in the area characterized by the highest noise level for all frequencies analysed.
It can be explained by location of this station close to the Kevitsa mine and the dam of a waterpower plant. Seismic noise
recorded by station P2 has the lower amplitude than the noise recorded by V1 station. It can be suggested that the main noise
sources at that station were the dam and the highway no. 4. A narrow peak at frequencies of about 26-30 Hz is seen in the
seismic noise spectrum of station V2. We suggest that the main noise source at the V2 was the dam. Unexpected results of
noise level estimations were obtained for stations P2 and A. As one can see in Figure 2, station P2 is characterized by relatively
high level of seismic noise. At the same time, seismic noise at station A is relatively low, despite similar distance from these
stations to the dam and to the road. We need to remember, however, that data acquisition along different profiles was made

during different time periods, so probably some additional high-frequency noise source was acting at P2 during the data acquisition period.

It is clear that dominating noise sources are different across the area of our study, and general condition for passive seismic interferometry (the sources need to be isotopically and homogeneously distributed around the study area) is not satisfied. However, local sources of high intensity can be used for evaluation of EGF for selected profiles. According to the results of spectral analysis and *a-priori* knowledge about locations of potential noise sources, the following possible candidates for sources of signals for passive seismic interferometry can be proposed:

1) Kevitsa mine, because all the profiles are located at distances of about 6 – 42 km from the mine.

2) Kitinen River and waterpower plants located on the river, because three of four profiles are located along the river.

3) The waves that are scattered on heterogeneities and can produce diffused wavefield, as proposed by van Manen et. al. (2005), Wapenaar (2004), Wapenaar et. al. (2015).

4) We can also use the signals from Vibroseis © and explosions recorded in XSoDEx refraction experiment, to utilize propagation of surface waves to long offsets. Such analysis is not possible with the data acquired in typical near-vertical reflection experiments, because only short offsets and limited recording times are used. In addition, active sources have relatively high frequencies, and they can be used only for shallow subsurface investigation.

In the next chapter, we investigate the wavefield produced by these possible sources using numerical modelling.

## 4 Numerical modelling of seismic wavefield from different sources

There is a few previous theoretical and numerical studies of the wavefield from various sources scattered on heterogeneities of elastic properties (Aki, 1969; Wu and Aki, 1985; Frankel and Clayton, 1986; Gritto et al., 1995; Bohlen et al., 2003). They showed that the scattered wavefield can be quite complicated, depending on the shape of heterogeneity and its elastic properties, location of the receiver in the far field or near field and other factors. For simulation of seismic wavefield propagation in the bedrock typical for XSoDEx area, we used SOFI3D software, which solves a wave equation by finite-difference method (https://git.scc.kit.edu/GPIAG-Software/SOFI3D/tree/Release). For simulation of the wavefield scattered on heterogeneities we developed synthetic model based on *a-priori* knowledge about geological structure of the study area. These main features are sub-vertically oriented mafic and ultra-mafic intrusions of irregular shape inside generally felsic bedrock composed of granites, gneisses and quarzites. In some places the bedrock is overlaid by quaternary sediments with thickness up to several dozen metres (Leväniemi et al., 2018; Karjalainen, 2019).

We used background velocities of Vp=5600 m/s, Vs=3500 m/s and density= 2650 kg/m$^3$ corresponding to felsic rocks. The embedded vertical high velocity bodies were representing mafic rocks intruded into the felsic rocks. The bodies were 30-150 m wide with depths varying randomly from 60 m to 600 m with the following physical properties (Figure 3): Vp = 6500 m/s, Vs =3700 m/s, density is 2800 kg/m$^3$. We also assumed an uppermost 60 m thick layers representing quaternary sediments with Vp=2000 m/s, Vs=1200 m/s and density is 1600 kg/m$^3$. The following elastic properties of air were used as boundary conditions of the model: Vp=330 m/s, Vs=0, density is 1.25 kg/m$^3$. As sources, we used: 1) plane waves from sources located

out of line in far field area, with frequencies of 50 Hz and 2.5 Hz and with an incidence angle of 45 degrees; 2) blast with dominant frequency of 30 Hz, located in the beginning of the profile; 3) waterpower plant (stationary noise with frequency of 2.5 Hz), located in-line with the profile. The assumption about major sources was made based on analysis of spectra, presented in Section 3, and our knowledge about locations of objects of human activities (Figure 2). In the modelling, we used the grid size of 30 m.

The first synthetic signal was a plane wave originating from a source in the far-field area. The wave front propagated from the depth of 6000 m with the angle of 40 degrees with respect to the profile direction and arrived at the surface at the incidence angle of 45 degrees. As one can see in synthetic seismogram in Figure 4, the recorded wavefield consists of the first arrival, several reflected waves and numerous scattered waves with apparent velocities of 2100-2500 m/s corresponding to surface waves. Figure 5 shows an example of particle motion diagrams (Figure 5c) and results of spectral analysis of these arrivals

(Figure 5b). Due to elliptical polarization and dependence of phase velocity on frequency, one can conclude that these scattered waves are surface Rayleigh waves. As these surface waves have stochastic directivities, superposition of them may be considered as diffused wavefield that can be used, in principle, to estimate EGFs.

In this synthetic example, we demonstrate that the diffuse wavefield consisting of low-frequency (5-20 Hz) surface waves (Rayleigh) can be produced by scattering of a high-frequency (50 Hz in our case) plane wave at velocity heterogeneities. We

considered monochromatic plane wave, but in real ambient noise many frequencies are usually present, thus the scattering would be more pronounced.

The second example simulates propagation of the signal originating from a production blast in the Kevitsa mine recorded by sensors of the Sakatti profile (Fig. 1). Figure 6 shows results of modelling of the wavefield produced by the blast and propagating in the model with stochastically distributed heterogeneities. The source function of the blast in this case is delta

function and the source is located in-line with the profile of seismic sensors. As seen, the wavefield consists of only direct P-wave arrival, reflected P-wave, multiples of P-waves, reflected from vertical heterogeneities, and surface Love wave. In that case, surface Rayleigh waves are absent from the wavefield. The scattered wavefield can be seen after approximately 2 sec at offsets of 2000-3600 m.

The third synthetic example corresponds to the direct wave continuously produced by waterpower plant, which could also be

scattered on heterogeneities and produce diffused wavefield. As positions of all the dams on Kitinen river are well known and all of them are located in-line with the Sakatti profile, we used the in-line position of the source in our simulation. As an input signal, we used a real seismic signal recorded by station V1 that was located at the shortest distance from the waterpower plant (Figure 2). The spectral-time diagram of the signal is presented in Figure 7. As one can see, there are several ranges of frequencies with some increasing of amplitudes (about 5 Hz, 12.5 Hz and 20-50 Hz). According to Antonovskaya et al. (2017;

2019), seismic noise generated by waterpower plants may correspond to a set of spectral peaks between 3.6 Hz and about 50 Hz. The other relatively high amplitudes could be due to production activities (transportation, excavation or others) at the Kevitsa mine. Therefore, in this case we have complex contribution of all these sources to the noise wavefield.

Figure 8 (a) shows synthetic seismograms of the stationary wavefield corresponding to the spectrogram presented in Figure 7. Analysis of a particle motion (Figure 8 (c)) shows that this stationary field is consisting of Rayleigh waves with apparent velocities of about 2100-2500 m/s. Figure 8(b) shows cross-correlations of the first trace with all other traces in Fig. 8(a). These cross-correlations show also P and S wave arrivals.

Results of our synthetic modelling demonstrate that the plane wave scattered at heterogeneities satisfies condition of diffuse wavefield and hence can be used to extract EGFs. The wavefield, produced by scattering of stationary signal from the waterpower plant can also be used in the cases when receivers are deployed within the first Fresnel volume area. Usage of the diffuse wavefield produced by scattering on local heterogeneities or of the stationary wavefield from a single source will have an advantage that in both cases the long registration time necessary for obtaining isotropic azimuthal coverage of ambient noise sources is not required. However, special analysis of the continuous data would be necessary, in order to extract the diffuse wavefield from the data. For this purpose, the SNRS algorithm described earlier in Afonin et al. (2019) can be used. The technique is based on the global optimization algorithm, in which the optimized objective function is a signal-to-noise ratio of an EGF, retrieved at each iteration. Maximizing the signal-to-noise ratio of the retrieved EGF is ensured by stacking only cross-correlation functions coherent with each other and corresponding to the stationary phase area. The details of the algorithm are provided in the Supplementary Material.

## 5 Verifying passive seismic interferometry with the scattered wavefield using passive seismic data recorded during XSoDEx experiment

To demonstrate application of passive seismic interferometry with the scattered diffuse wavefield, we used passive seismic data acquired in the XSoDEx experiment. We applied the SNRS algorithm for diffuse field extraction and for the EGFs evaluation. From the XSoDEx lines, one is particularly suitable for such demonstration. In this short high-resolution profile (green line in Fig. 9) of total length of 1000 m both 1C and 3C Sercel wireless units were installed at distances of 10 m. The 3C sensors were installed between 1C sensors at distances varying from 20 to 30 m. The results of passive seismic interferometry along this line can be also verified using the active source seismic data acquired along the same line and results of previous geophysical experiments and drilling in this area.

For retrieving of EGFs and further dispersion curve calculation, we used continuous passive seismic data recorded during the period of 21.08.2017 – 23.08.2017 (about 48 hours). The workflow for data analysis used in our paper is shown in Fig. 10. We made no *a-priori* assumptions about the nature and spatial distribution of noise sources. For calculation of EGFs, we applied such pre-processing procedures as removing mean and trend, spectral whitening and prefiltering by the band pass filter of 1-100 Hz. After this, we calculated cross-correlation functions in such a way, that virtual sources of impulse signal were placed in the beginning, in the middle and in the end of the profile. The choice of virtual source positions is based on the dominant wavelength of the analysed signal that is about 400 m. An example of EGFs calculated with the SNRS algorithm and the correspondent dispersion curve are presented in Figure 11.

As seen from particle motion diagram calculated for one of the 3C sensors, the main arrival seen in EGFs corresponds to Rayleigh wave. A good coherence of waveforms of dispersed surface wave is also seen.

For calculation of velocity models from these three dispersion curves, we used Geopsy software (www.geopsy.org). We applied global optimization algorithm with 500 iterations to obtain the best solution. The starting model consisted of three major layers with properties presented in Table 1. The range of each property was selected using information about physical

properties of rocks in the study area available from literature (Dortman, 1992; Leväniemi, 2018; Schön, 2015).

Table 1 – Starting model of the medium, used for inversion of dispersion curves.

| Depth | Vp, m/s | Vs, m/s | Rho, kg/m$^3$ | rock types |
|---|---|---|---|---|
| 1-50 | 700-1200 | 350-900 | 1200-1500 | quaternary deposits, coarse-grained sorted sediments |
| 50-200 | 5900-6200 | 1800-3600 | 2000-2300 | granite, felsic volcanic rocks |
| 200-500 | 6300-6600 | 3300-3600 | 2000-2500 | mafic volcanic rocks |

We calculated 1D velocity models for each 100 m of the profile, using position of virtual sources at 0 m, 450 m and 900 m

(Figure 11 (a)). After that, using triangular and linear interpolation of 1D models, we obtained a 2D model presented on Figure 12(a). The subplots 12(c) and 12(d) present results of global optimization of one selected dispersion curve. The 2D velocity model in which possible rock types are indicated by different colours, presented on subplot 12(b). The ranges for S-wave velocities are defined according to (Dortman, 1992). One interesting feature of this model is a layer with S-wave velocities of about 200-400 m/s and thickness of 20-38 m that may correspond to sediments. The thickness of sediments agrees well with

the result of Åberg et al. (2017), who interpolated results of GPR measurements and drilling information to obtain sedimentary thickness in this area. According to them, the sedimentary thickness in the studied area is about 25-30 m. The velocities of S-waves beneath the sedimentary cover will be discussed in Section 6.

We applied the same technique of EGFs calculation to the data with lower spatial resolution recorded along the part of Sakatti profile shown by blue in Figure 9. Particle motion diagram (Figure 13(c)) shows that evaluated EGFs contain mainly surface

waves. Dispersion curves were calculated for each 500 m of the profile for obtaining 2D velocity model. For this, the virtual sources were placed at each 500 m and cross-correlation functions were calculated between the virtual source receiver and all receivers located at distances no larger than 1000 m from the virtual source. For calculation of dispersion curves, the Multichannel Analysis of Surface Waves (MASW) technique was used.

As one can see on Figure 13, the passive seismic interferometry with the SNSR allowed us to evaluate dispersion curve for

frequencies of about 3.5-7 Hz. Inversion of dispersion curves was used to obtain 1D velocity models that were combined into a 2D model (Figure 14(a)).

For verification of the modelling results, we compared the velocity model in Figure 14(a) with the model obtained by inversion of dispersion curves estimated from surface waves produced by scattering of signal from the controlled source for the same part of Sakatti line (Figure 14 (b)). We compared velocity models, because they are obtained at the last step of data processing, where all the errors from previous steps are accumulated. In addition, we noticed that the differences between EGFs and dispersion curves for these two data sets are insignificant. As seen, the velocity models reveal the same details, and the velocities are generally in good agreement. From Figure 13 (b) one can see that the width of error bars of dispersion curves are about 500 m/s. Differences in velocities between two 2D models (Figure 14) are within these limits. The differences in velocities are of the order of 100 m/s in the central part of the profile. The largest difference up to 600 m/s can be seen in the beginning of the profile (from 15000 to 15500 m) and it can be explained either by uncertainty in dispersion curves extraction or by inversion errors.

## 6 Shear-wave velocity models obtained using Vibroseis © signal scattered at heterogeneities

Surface waves recorded in active source experiments (ground roll) usually considered as non-wanted signal and removed from the data during processing. However, S-wave velocity models can be obtained from Vibroseis© surface waves using MASW method (Al-Husseini et al., 1981; Mari, 1984; Gabriels et al., 1987; Park et al., 1999). In this case, the depth resolution for S-wave velocity models is limited to several meters due to short offsets and small registration time in near-vertical reflection data. As the data in XSoDEx experiment was recorded at long offsets with the wireless equipment, such recordings can be used to obtain the S-wave velocities at larger depths.

Examples of raw shot gathers of Vibroseis© signals recorded in reflection experiment are presented in Buske et al. (2019). They used bandpass filters of 30-40-100-120 Hz to eliminate surface waves from raw reflection data. The sweep frequencies were from 10 to 170 Hz. In raw reflection data the surface waves arrivals can be followed up to 2-3 sec at rather short offsets of about 350 - 400 m. An example of record section compiled from wireless recorders data deployed at 160 m interstation distances is presented in Figure 15. In this section a vibrator signal, presented in Figure 16, was correlated with the traces recorded at long offsets. In Figure 15, one can see the first arrival of P-wave with velocity of about 5400 m/s and direct Rayleigh wave with velocity of about 880 m/s in frequency band of 20–100 Hz (Figure 15 (b)) that can be followed to offsets of about 1 km. In the frequency band of 1-10 Hz (Figure 15 (a)), the surface waves cannot be followed to long offsets. The surface wave arrivals that can be correlated appear on several traces at short offsets. As seen in Figure 16 (b), the frequencies of the vibrator signal start from about 12 Hz and no lower frequencies are present.

We used the SNRS technique and continuous seismic recordings of XSoDEx experiment to obtain EGFs for all the XSoDEx profiles. As seen in Fig. 17 (a), the EGFs cannot be extracted from our data using conventional passive seismic interferometry, where simple stacking of cross-correlation functions is used (Shapiro and Campillo, 2004; Bensen et. al., 2007; Wapenaar et. al., 2011; etc.). In all EGFs the main phase seen is Rayleigh wave (Figure 17 (b)). The EGFs were used to obtain dispersion curves and invert them using Geopsy inversion software and model parameters, presented in Table 1. Figure 18 shows the 2D velocity models for all XSoDEx profiles to the depth of 300 m obtained by interpolation of 1D velocity models. The same

velocity models, in which different colours indicate possible rock types, are presented on Figure 19. The ranges of S-wave velocities correspond to major rock types of the Fennoscandian shield (Dortman, 1992). The boundaries of major lithological units from Figure 1 are also indicated on Figure 19. All the velocity models are shown in the same colour scale.

The S-wave velocities along the XSoDEx profiles are generally varying from very low values of 200-400 m/s detected in some places up to 3200 m/s. The uppermost layer with velocities of 200-400 m/s and with thickness up to 50 m corresponds to quaternary sediments, and the boundary between this layer and the lowermost part of velocity models is indicated also by the velocity contrast in 1D models. Independent information about thickness of sediments in our study area obtained by direct drilling (summarised in Karjalainen, 2019) shows that the thickness of quaternary sediments there is no more than 40-50 meters. That is why it can be concluded that the S-wave velocities in the range of 800-3200 m/s correspond to different types of basement rocks. As these values are generally much lower than the values of S-wave velocities for the rocks of the Fennoscandian Shield, obtained by laboratory measurements on rock samples (Kern et al., 1993, Dortman, 1992), they cannot be interpreted directly in terms of rock composition.

As known, non-zero azimuths to noise sources may only increase apparent velocities (e.g. Sadeghisorkhani et al., 2016). Therefore, it is necessary to find another explanation of generally low S-wave velocities of the basement rock in our study.

The S-wave velocities in the uppermost crust down to several kilometres were previously evaluated using surface waves by Pedersen and Campillo (1991), Grad and Luosto (1992), Grad et al. (1998). They reveal low values of S-wave velocity in the shallow crust and low values of quality factor that is rapidly increasing at the depth of about 1 km. Moreover, Grad et al. (1998) found that the quality factor (Q-factor) in the Archean shallow crust of the Fennoscandian Schield is lower than that in the Proterozoic crust. Grad and Luosto (1992) explained the low Q-factor in the uppermost 1 km of the crust by increased cracks density. Therefore, increased crack density can explain also generally low S-wave velocities of the basement rocks revealed by our study. Consider that the wave is propagating through fractured medium consisting of granites and the fractures are filled with some clastic rocks. If the S-wave velocities in the granitic rock are about 3300 m/s and those in the clastic rocks are about 400 m/s (like the velocity in quaternary deposits), then the averaged velocity would be about 1700-2000 m/s, depending on crack density. Taking the general effect of fracturing into account, we conclude that the values of S-wave velocities lower than 2000 m/s correspond to felsic rock, while the higher values of velocity correspond to mafic and ultramafic rocks (Figure 18).

# 7 Conclusions

In our study, we used the data of ambient noise recorded during short time period in a generally quiet area with low level of anthropogenic noise. We showed that the noise was non-stationary, and that the azimuthal distribution of noise sources was neither isotropic nor heterogeneous during the whole data acquisition period. In spite of that, we obtained good quality EGFs, dispersion curves and the S-wave velocity models showing presence and thickness of quaternary sedimentary cover and velocity heterogeneities in the bedrock that agree well with the geological data. We explain this by the fact that the ambient noise recorded during the XSoDEx experiment contained a large proportion of diffused wavefield produced by scattering of plane waves from distant sources and ballistic waves produced by certain types of non-stationary sources on stochastically

distributed heterogeneities in the uppermost crust. We demonstrated by numerical modelling that certain types of geological structures, particularly those composed of rocks with contrasting elastic properties, could scatter plane waves and ballistic

waves from non-stationary sources and produce scattered Rayleigh waves. This scattered wavefield together with the ballistic wavefield from non-stationary sources can be used for EGFs evaluation, if the special algorithms are applied. One opportunity is to use signal-to-noise ratio stacking (SNRS) algorithm. Our result is a practical illustration of the conclusions about retrieval of EGF from the scattered wavefield revealed previously by Wapenaar (2004) and Wapenaar and Thorbecke (2013). In certain geological areas, extraction of EGFs from the wavefield scattered at heterogeneities, provides an opportunity to reduce the

time for short-term passive seismic experiments.

## 8 Acknowledgements

The XSoDEx project was realised as a joint effort of Geological Survey of Finland (coordinator), TU Bergakademie Freiberg, Institute of Geophysics and Geoinformatics, Freiberg, Germany, and University of Oulu. The wireless equipment of the University of Oulu is jointly operated by Oulu Mining School and Sodankylä Geophysical Observatory of the University of

335 Oulu. The field work of the University of Oulu personnel in summer, 2017 was supported by the Renlund Foundation. Financial support for processing and interpretation of the XSoDEx data used in this study was provided by Geological Survey of Finland in 2018. Thanks to Sodankylä Geophysical Observatory staff for their kind assistance during XSoDEx survey. Particular thanks are to Hanna Silvennoinen, Jouni Nevalainen, Kari Moisio, Jari Karjalainen and Tommi Pirttisalo, whose participation in XSoDEx field survey was particularly important for safe and reliable operation of Sercel Ltd wireless equipment. Many thanks

to Henrik Jänkävaara for his contribution in XSoDEx data processing.

The authors wish to acknowledge CSC – IT Center for Science, Finland, for computational resources. The numerical modelling of wavefields, presented in the work, was funded by the Russian Federation Ministry of Science and Higher Education according to the research project AAAA-A18-118012490072-7.

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

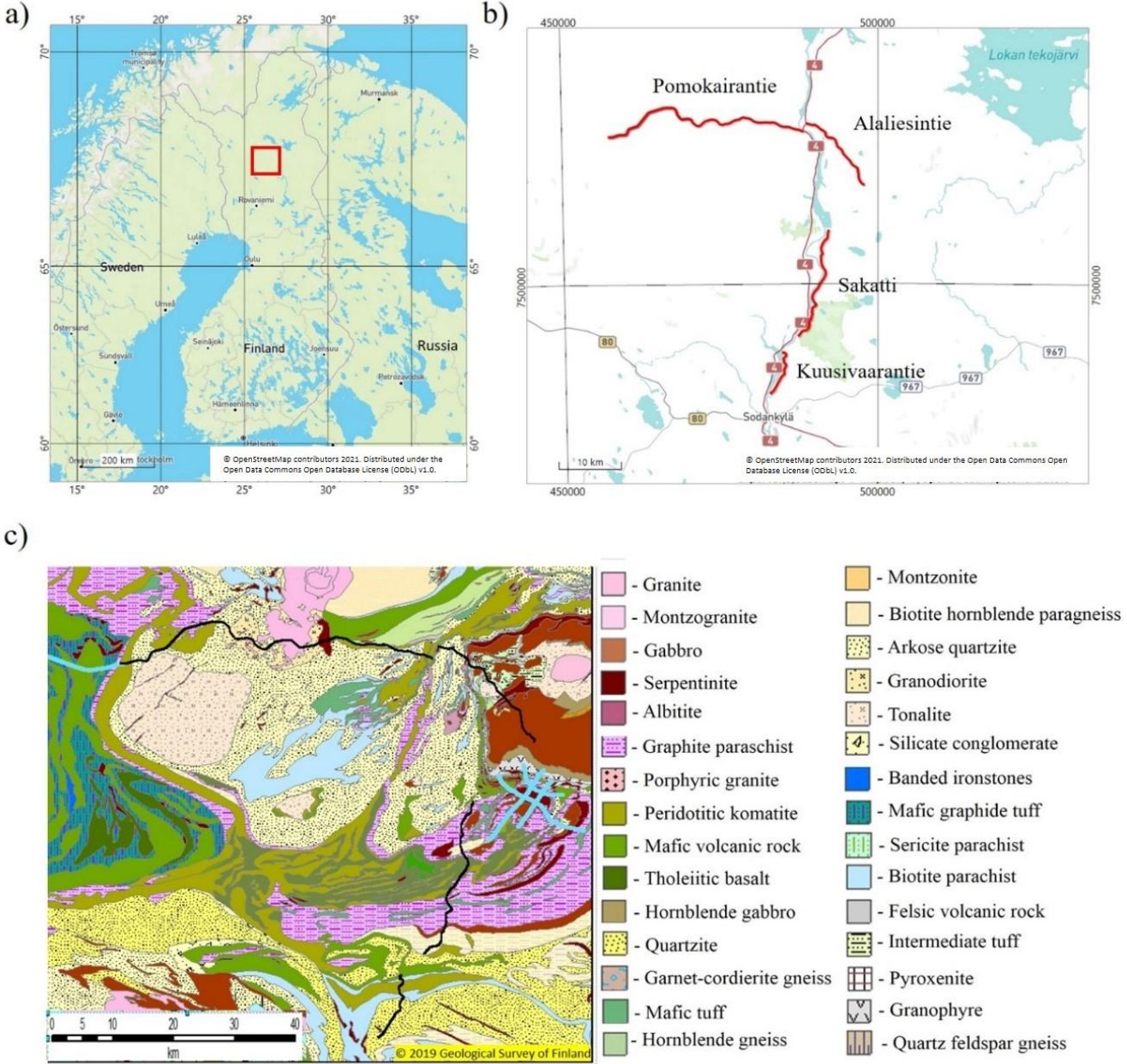


**Figure 1: The XSodEx seismic experiment: a) geographical location of the study area (red rectangle); b) the XSoDEx survey lines (coordinate system EUREF FIN TM35FIN); c) geological map of XSoDEx study area (Buske et. al., 2019). The XSoDEx survey lines are shown with black lines. Previous seismic reflection survey lines are plotted with blue lines.**

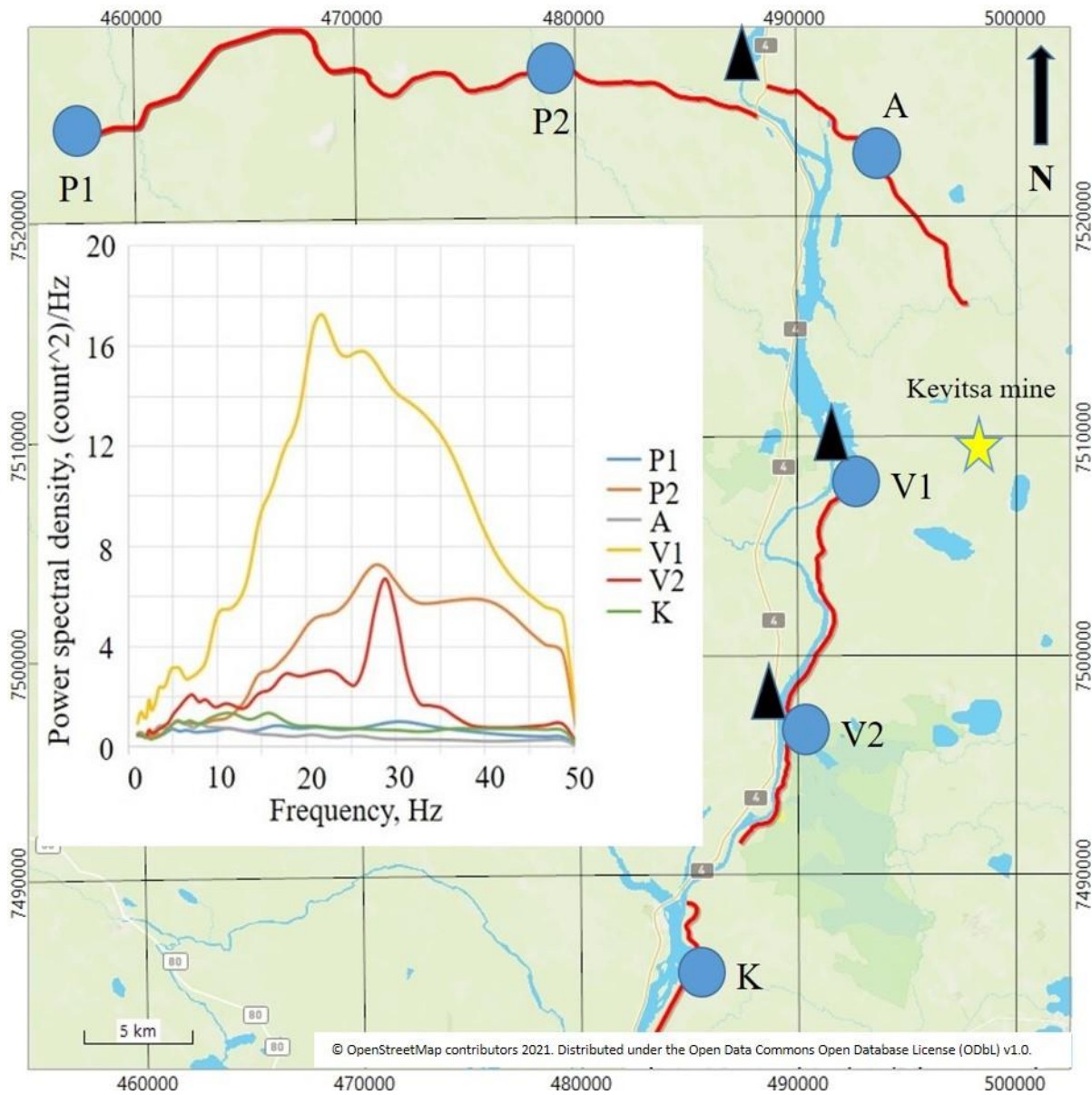

**Figure 2: Illustration of noise level at six stations located in different profiles of the XSoDEx experiment. Position of strong noise sources (Kevitsa mine (yellow star) and dams of waterpower stations (black triangles)) and seismic stations selected for analysis (blue circles) are indicated. Brown and white lines show roads. Inset figure shows comparison of ambient noise power spectral density estimated at selected stations.**

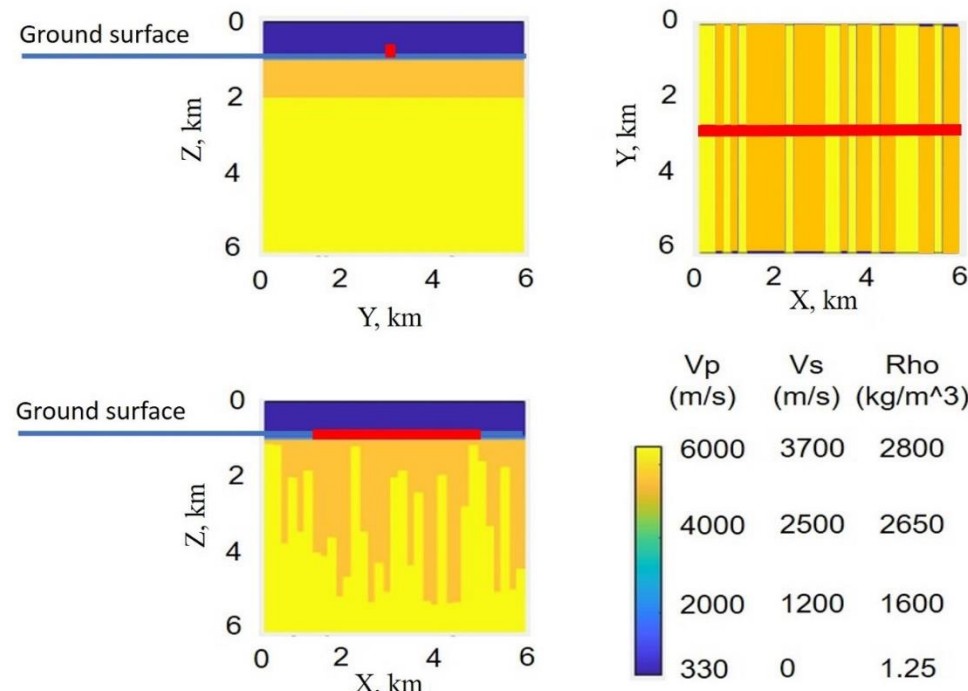

**Figure 3: The synthetic model used for investigation of wave propagation. Position of the seismic profile marked by the red line. Low velocity corresponding to the air is indicated on the top by blue colour.**

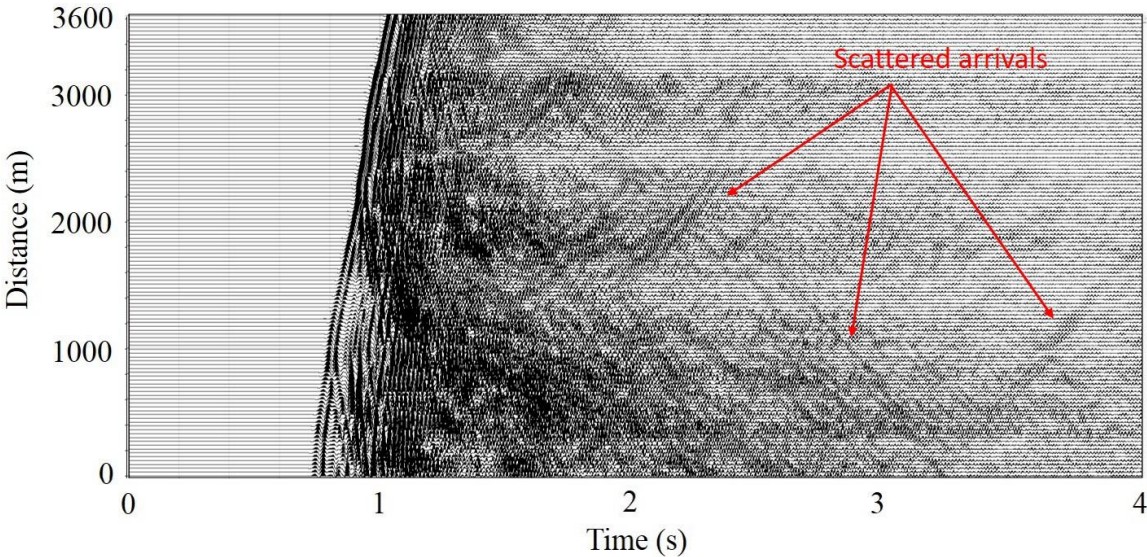

**Figure 4: An example of synthetic seismogram (vertical component) of plane wave arrived with incidence angle of 45 degree and propagated through the synthetic model (0-4 sec). The seismogram shows first arrivals and numerous reflections. From about 1.5**
**seconds we can see scattered arrivals of different directions with apparent velocities of 2100 -2500 m/s. Several arrivals of scattered waves are indicated by red arrows.**

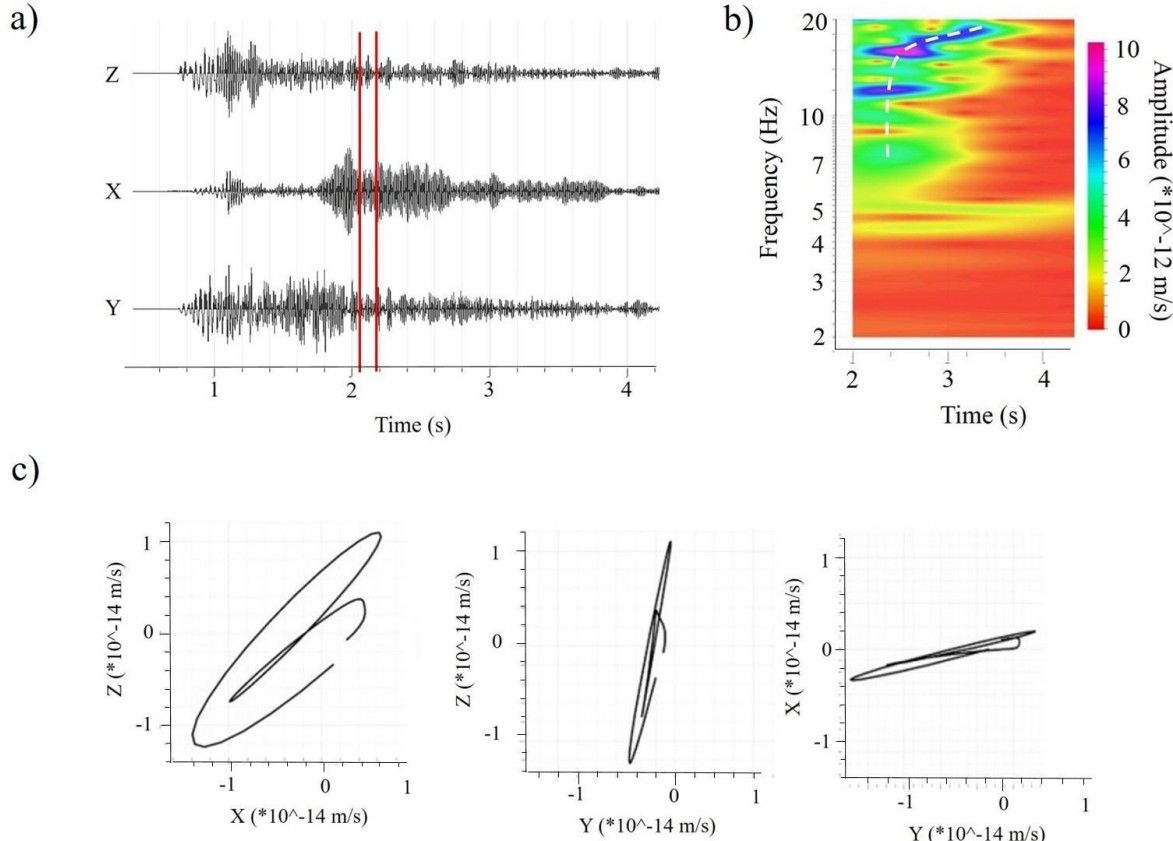

**Figure 5: Result of analysis of scattered arrivals in synthetic seismograms with apparent velocity of 2100 -2500 m/s for frequencies 2-20 Hz (Fig. 4): a) 3C seismogram for synthetic receiver located at distance of 2000 m from the source (Fig. 4); b) spectrogram of the scattered arrival, recorded by vertical component of synthetic receiver (white dashed line illustrate dispersion properties of the scattered arrival); c) particle motion diagrams, calculated using part of seismogram indicated by red lines in Figure 5 (a), which corresponds to scattered arrival.**

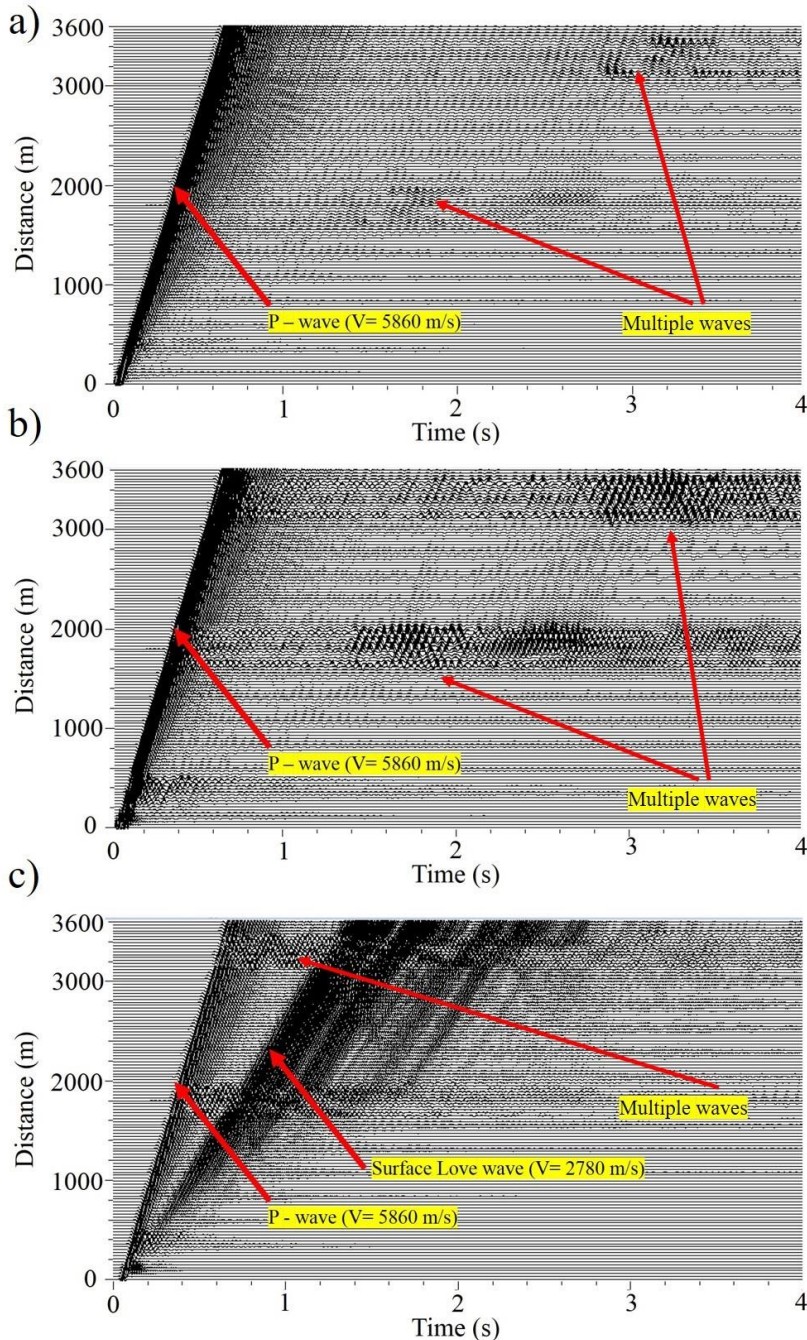

**Figure 6: Synthetic seismograms of the blast. The source was located at offset of -300 m: a) vertical channel Z; b) horizontal channel X; c) horizontal channel Y.**

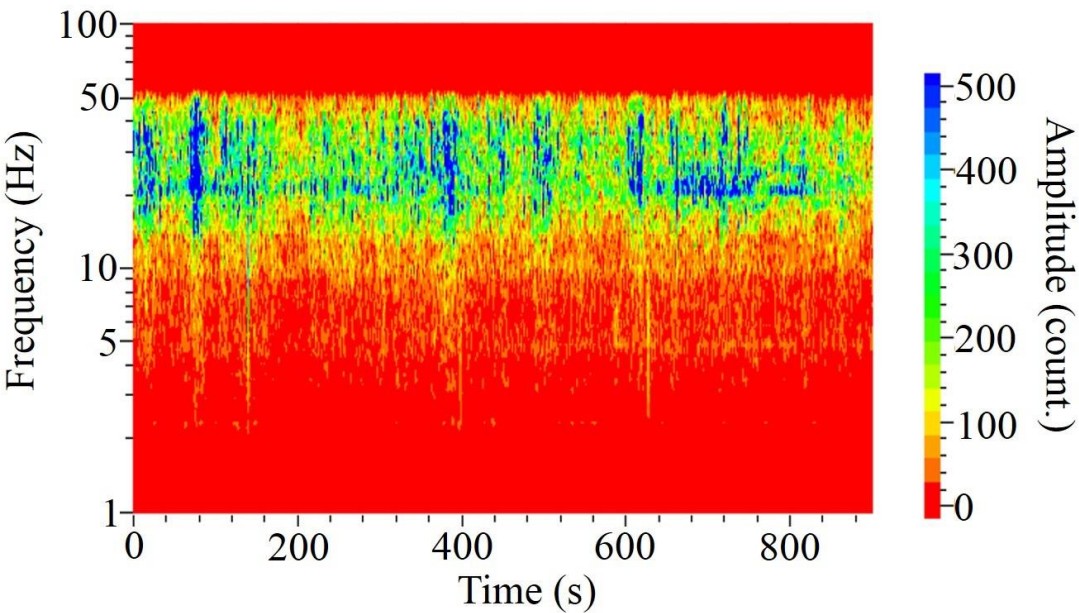

**Figure 7: Spectrogram of the signal recorded by station V1 (location is shown in Figure 2) and used as a wavefield of the source in simulation.**

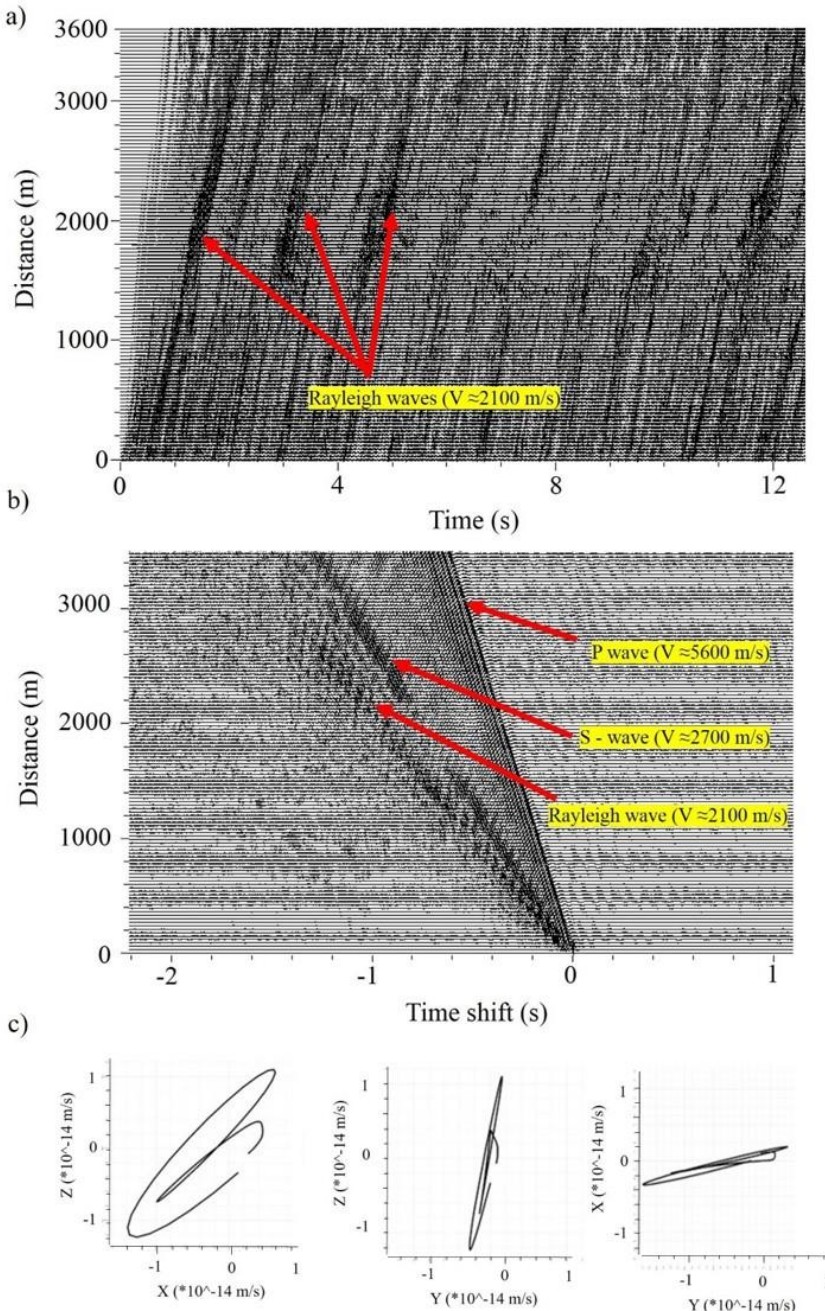

Figure 8: Synthetic seismograms of the signal with the spectrogram presented in Fig. 7 and propagating in synthetic model shown in Fig. 3. The virtual source is located at offset of 0 m: a) seismograms of vertical components; b) cross-correlation functions of vertical components, calculated between the first trace and all other traces in Figure 8 (a); c) particle motion diagram of Rayleigh wave part in Figure 8 (b).

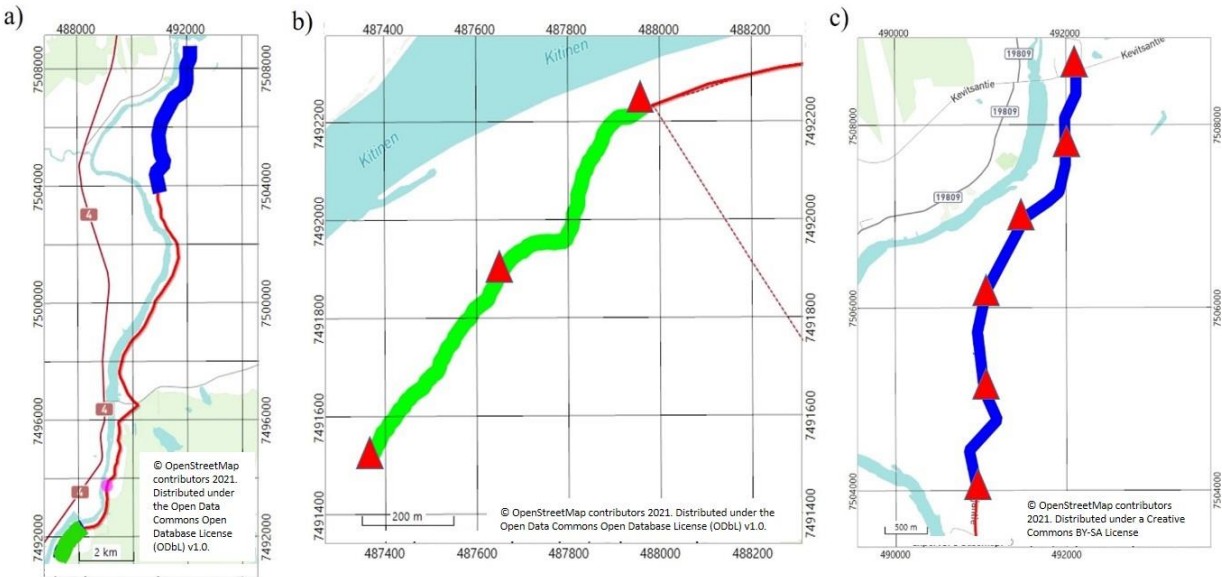

Figure 9: Location of passive seismic profiles of XSoDEx Sakatti line: a) positions of high-resolution line (green color) and profile of lower resolution (blue line); b) The high-resolution line in which red triangles marks positions of virtual sources; c) profile of lower resolution (red triangles indicate positions of virtual sources).

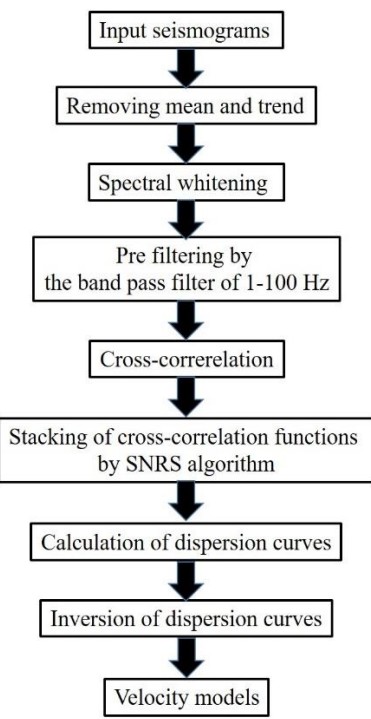

Figure 10: The workflow for data analysis

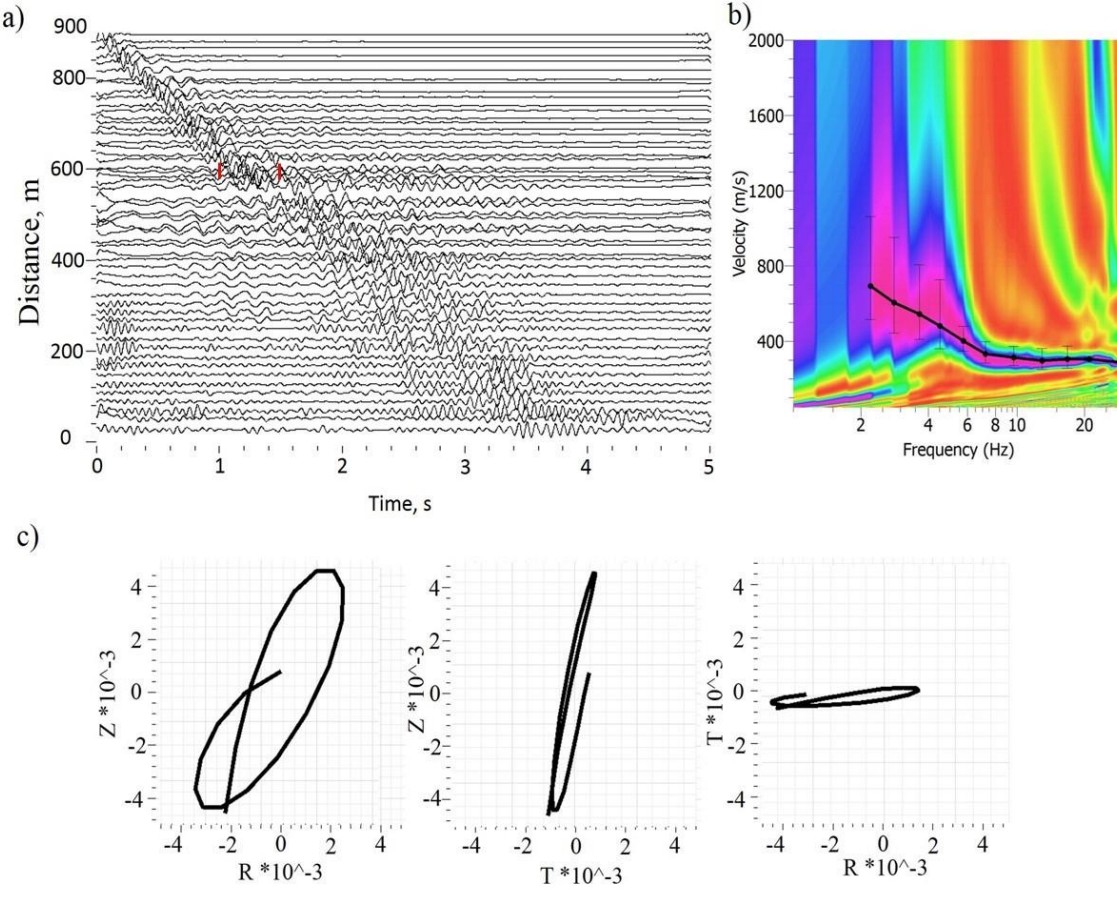

**Figure 11: Example of EGFs with the correspondent dispersion curve, obtained from passive seismic data for high-resolution profile shown in Fig. 9: a) EGFs; b) dispersion curve, extracted by MASW technique; c) particle motion diagrams for part of EGF indicated in (a) by red lines.**

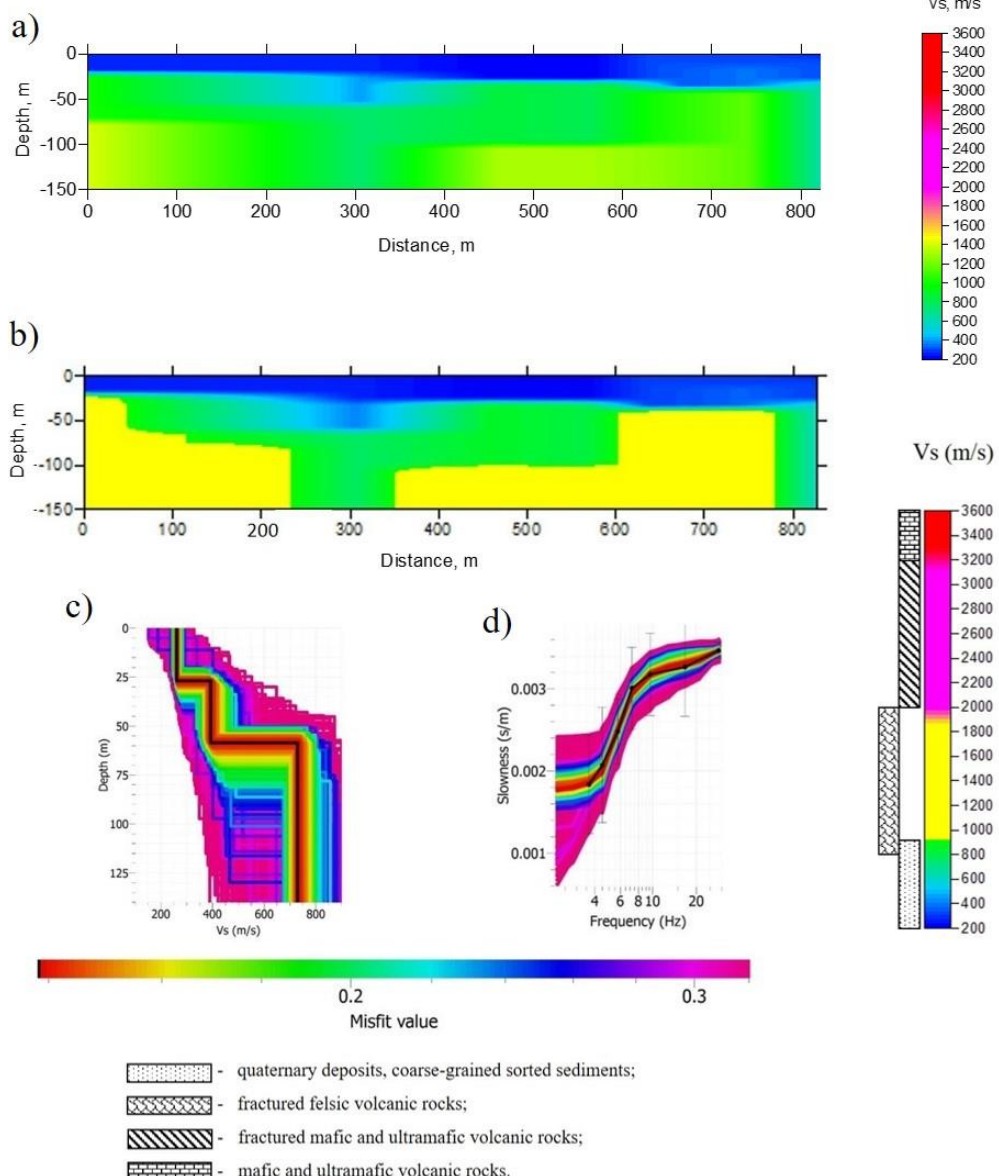


**Figure 12: Results of inversion of dispersion curves obtained from passive seismic data of high-resolution XSoDEx profile: a) 2D velocity model obtained by interpolation of 1D velocity models. Velocity colour scale is shown in the left. b) 2D velocity model in which possible rock types are indicated by different colours. The ranges for S-wave velocities are defined according to (Dortman, 1992), see also Figure 19. Velocity colour scale is shown in the left. c) an example of 1D velocity model (black line) obtained by**

**inversion of the dispersion curve in (d), corresponding to distance of 200 m in Fig. 12 (a); d) dispersion curve (black line), used for inversion of the velocity model in Fig. 12 (c). Colour scale corresponds to values of misfit function during different iteration steps in global optimization algorithm.**

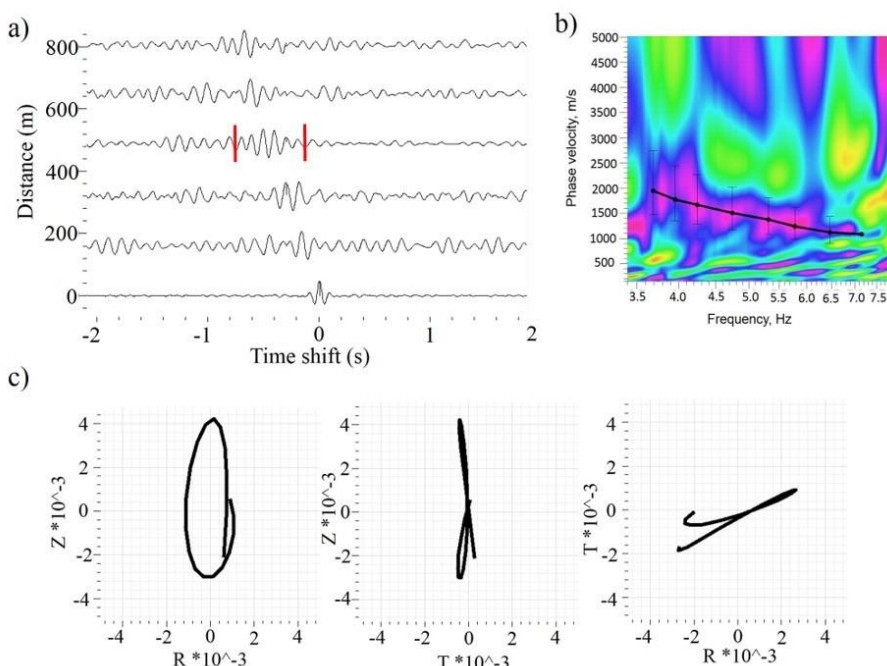

**Figure 13: Example of EGFs with correspondent dispersion curve, obtained from passive seismic data: a) EGFs on frequency band 3-8 Hz; b) dispersion curve, calculated from EGFs by MASW technique; c) particle motion diagrams of a surface wave part of an EGF indicated by red lines.**

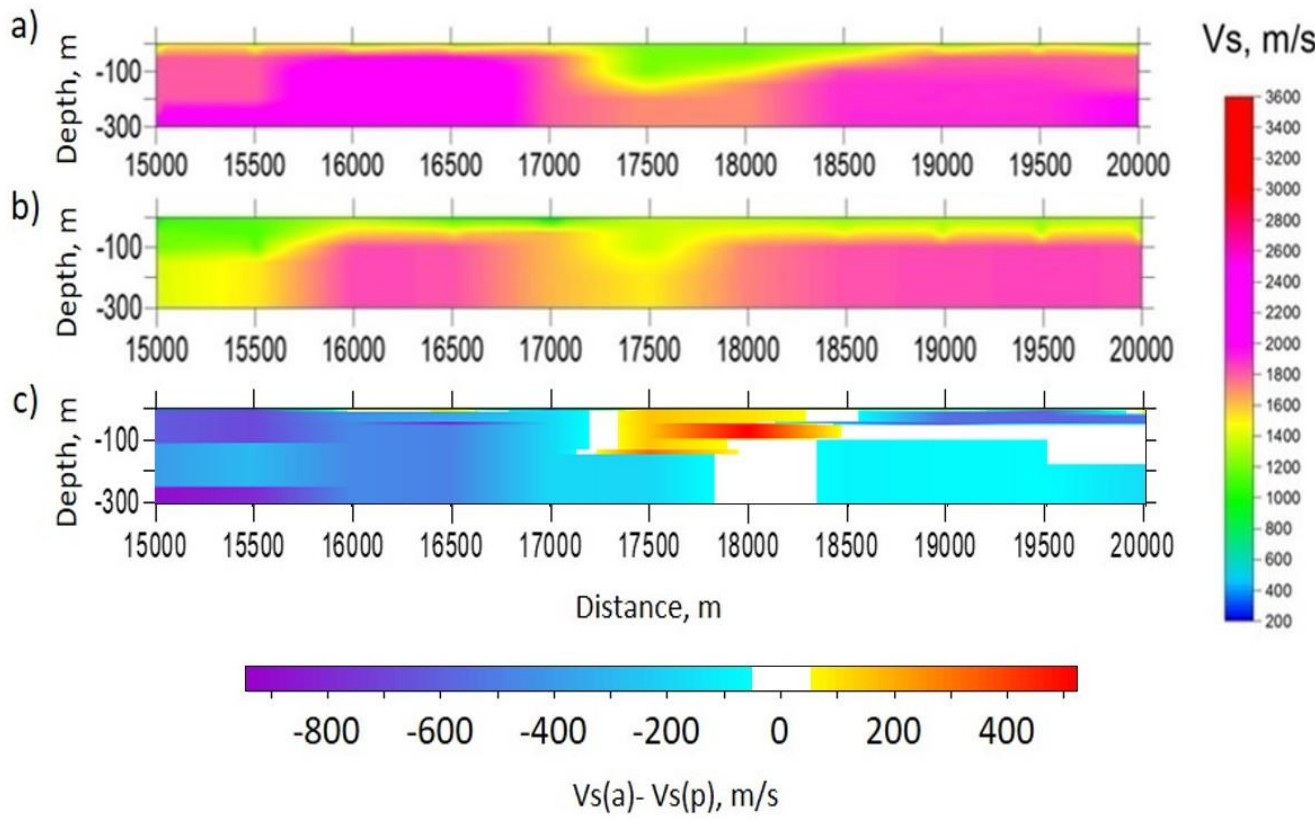

**Figure 14: Velocity models calculated by inversion of dispersion curves along the part of Sakatti profile marked by blue in Fig. 9. (a) S-wave velocity model obtained using passive seismic data; (b) S-wave velocity model obtained using seismic data, which contain signals produced by controlled source; c) differences between S-wave velocity models in plots (a) and (b).**

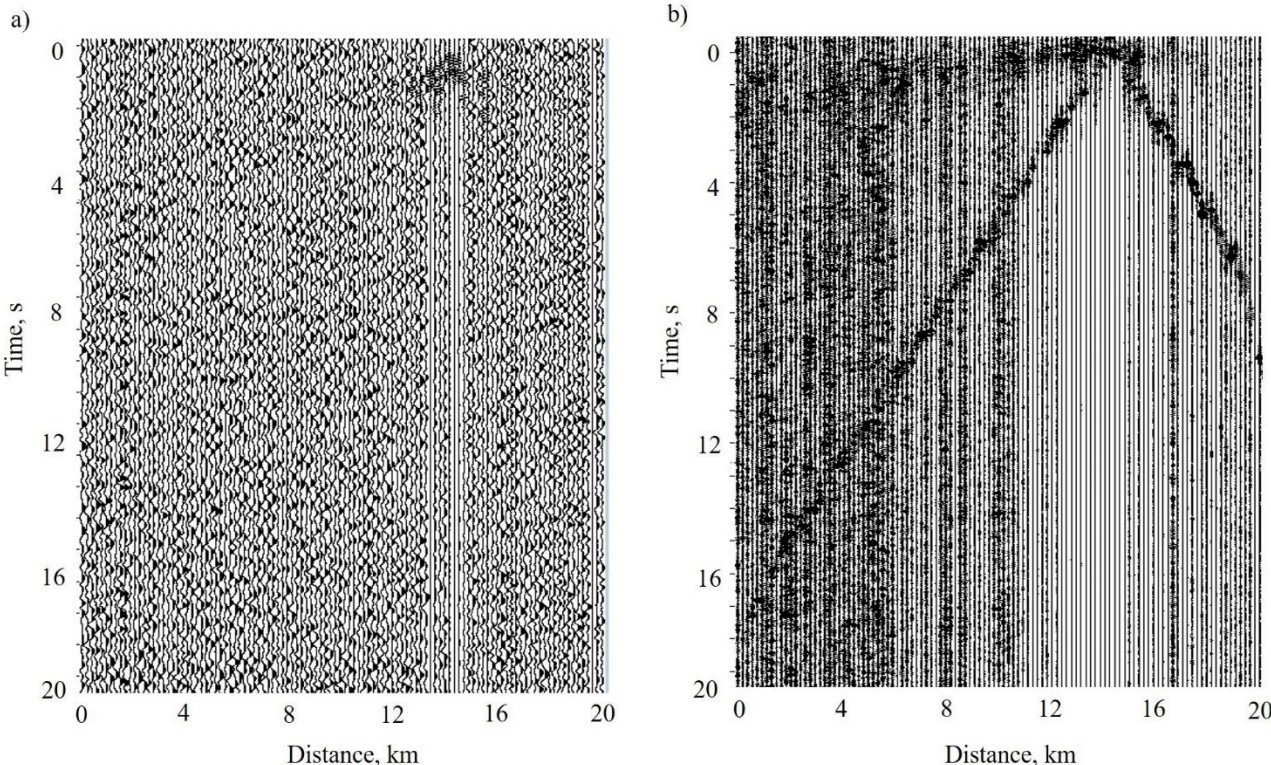

**Figure 15: Cross-correlation functions of signals, produced by Vibroseis© with the data recorded by wireless equipment at large offsets in frequency band of: a) 1-10 Hz, amplitudes normalized at maximum of each trace; b) 20-100 Hz. The signal used for cross-correlation is shown in Fig. 16.**

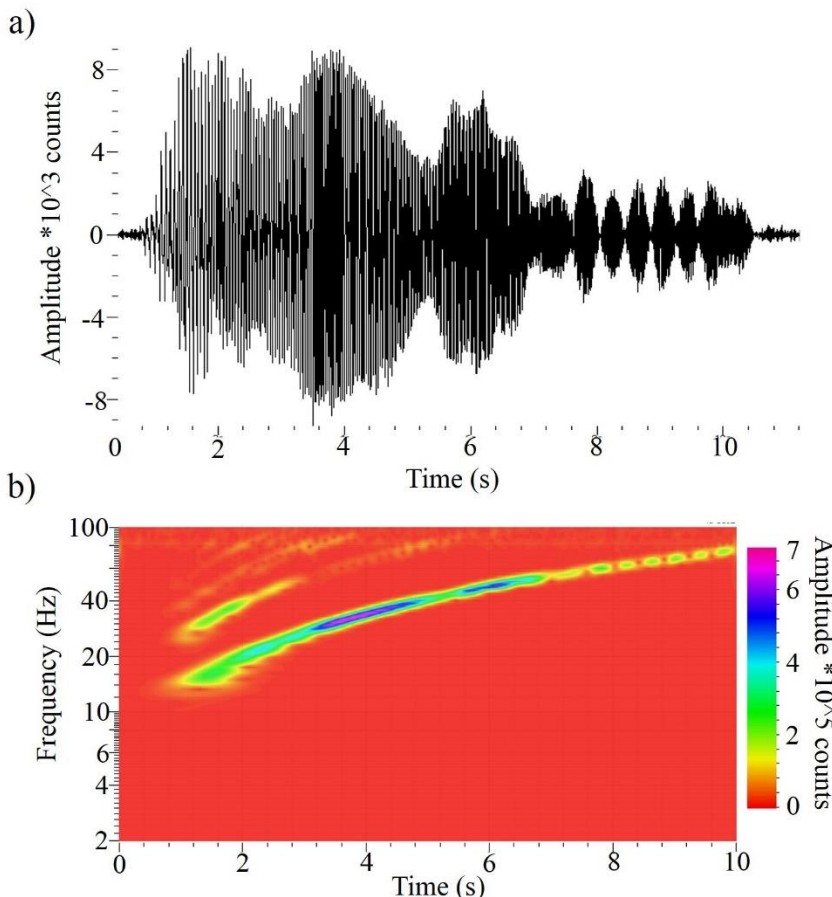

**Figure 16: An example of Vibroseis© sweep, recorded by a wireless sensor placed near the vibrator in XSoDEx experiment: a) seismogram; b) spectrogram showing the frequency content of a vibrator signal.**


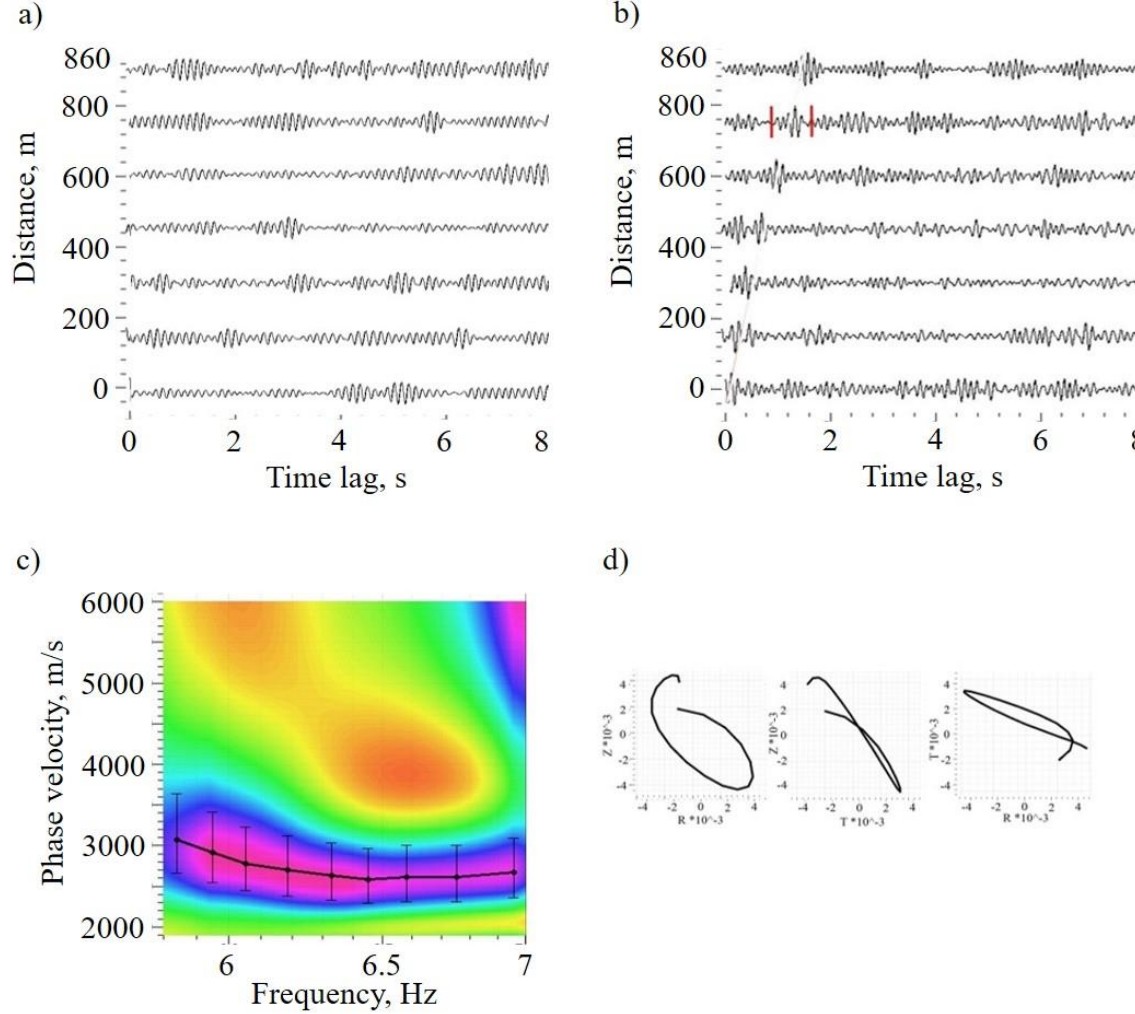

**Figure 17: An example of EGFs with correspondent dispersion curve of Rayleigh wave on frequency band 5-10 Hz, obtained by passive seismic interferometry: a) EGFs, obtained by conventional method; b) EGFs, obtained by passive seismic interferometry with SNRS algorithm; d) dispersion curve, extracted by MASW technique; c) particle motion diagrams for surface wave part of EGF marked by red in (a).**


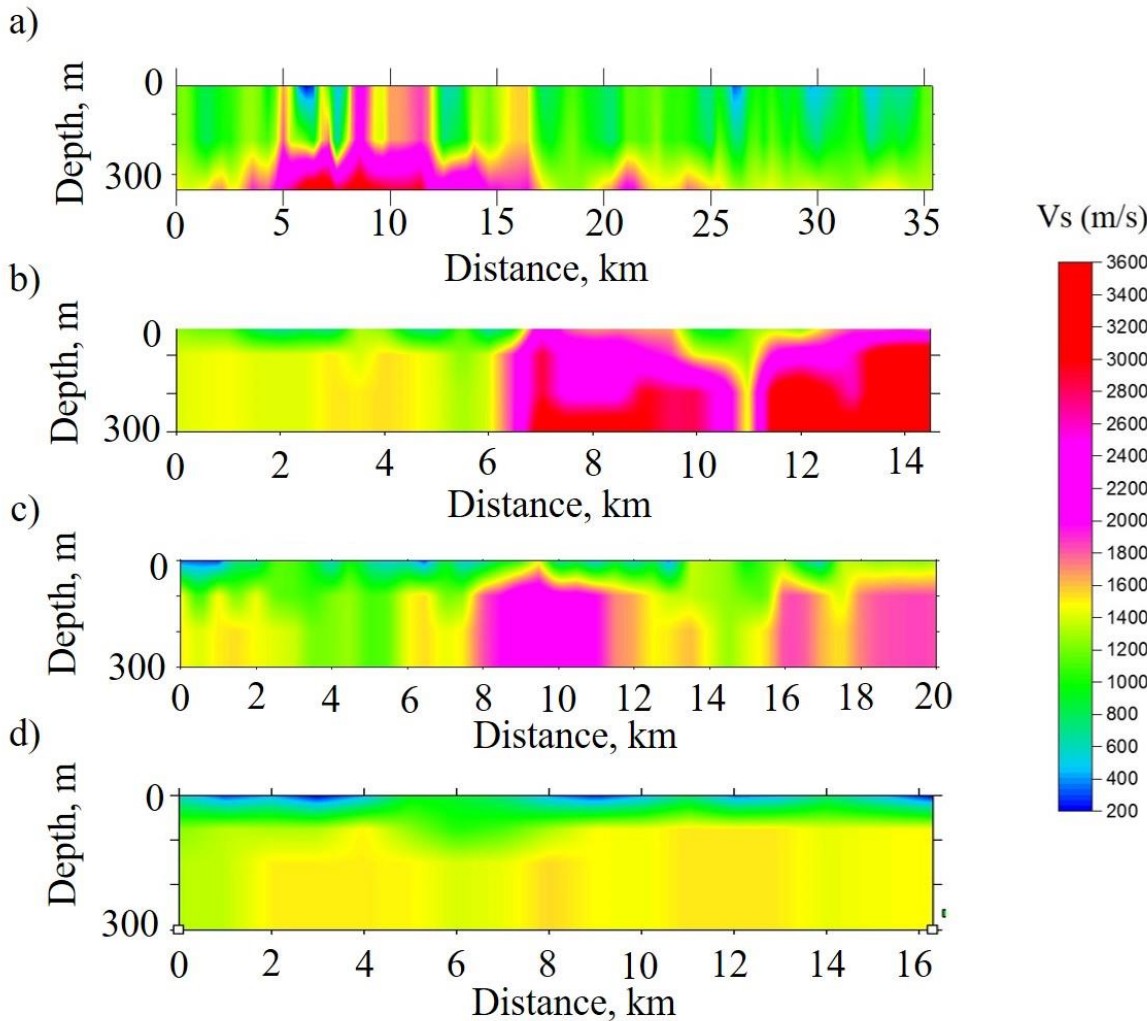

**Figure 18: Velocity models, calculated by inversion of dispersion curves, obtained by passive seismic interferometry for XSoDEx profiles shown in Fig. 1. a) Pomokairantie; b) Alaliesintie; c) Sakatti; d) Kuusivaarantie.**

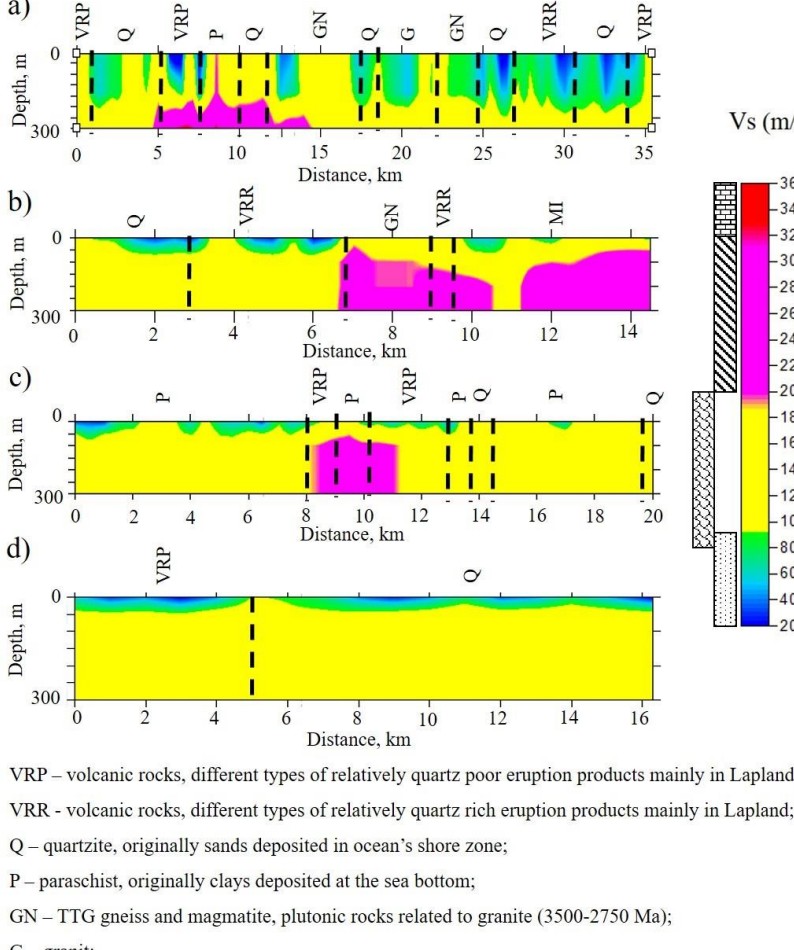

VRP – volcanic rocks, different types of relatively quartz poor eruption products mainly in Lapland;

VRR - volcanic rocks, different types of relatively quartz rich eruption products mainly in Lapland;

Q – quartzite, originally sands deposited in ocean's shore zone;

P – paraschist, originally clays deposited at the sea bottom;

GN – TTG gneiss and magmatite, plutonic rocks related to granite (3500-2750 Ma);

G – granit;

MI – layered mafic intrusions, Tornio-Koillismaa belt (2440 Ma).

- quaternary deposits, coarse-grained sorted sediments;

- fractured felsic volcanic rocks;

- fractured mafic and ultramafic volcanic rocks;

- mafic and ultramafic volcanic rocks.


**Figure 19: Velocity models presented in Fig.18, in which possible rock types are indicated by different colours. The ranges of S-wave velocities correspond to major rock types of the Fennoscandian shield (Dortman, 1992): a) Pomokairantie; b) Alaliesintie; c) Sakatti; d) Kuusivaarantie.**


