# Peer review of "Near surface structure of Sodankylä area in Finland, obtained by passive seismic interferometry"

_Solid Earth, 2020_

## Referee Comment (RC1) · Yunhuo Zhang (Referee) · 17 Nov 2020

I have gone through the entire pre-print. It is an interesting study and useful reference. It can be considered for publication, provided some areas can be improved. Please refer below for your consideration: The title of the paper highlights the 'advanced method', which is the SNRS algorithm to estimate the green's function from diffusive ambient noise field. However, the SNRS is just referred to the author's earlier paper, without any elaboration. This makes the title not reflects the content correspondingly. Since the SNRS algorithm is already published and discussed earlier, it is suggested to amend the title accordingly, either highlighting the case study, or the

attempts to characterize the noise field of the sites, etc. Section 4, several synthetic models are created to characterize the noise field of the site, taking into consideration of the nearby major activities. It is quite interesting and worth expanding. The assumptions of major sources need more explanation. The key message of creating the synthetic models are to support the claim that the noise field of site is diffusive. However, it is very common in elsewhere, too. Therefore, it would be better to draw some more novel conclusions from the synthetic models. Noted that each synthetic model is to simulate one type of sources. Would it better to create an overall model that combines all possible noise sources. If this one can be done, the authors may explore full waveform inversion of passive seismic waves. It is essential to beef up the field acquisition in more and clearer details, e.g., the field plan out, geophone type and corner frequency, sampling rate, source signature and location for active testing, etc. Figure 1 is not clear where are the blue/black lines. It is also difficult for the readers who are not familiar with Finland geology without necessary introduction. The quality of Figure 2 needs to be improved to meet the criteria of publication. It is not clear about the caption of Figure 5 (a) that what is the distance of 2000m referring to. Figure 5(b) horizontal axis and color bar scale seem not correct, if it is a dispersion image. Figure 6 (b) and 6 (c) are very interesting. It is worth expanding the explanation why these 2 directions are so different, whereby 6(b) can't see surface wave and 6 (c) can see surface wave clearly. Figure 7 shows the source is mainly in 10-40 Hz, which is quite high. Please explain how such high frequency source can illuminate to 300m below ground. Figure 11 (a), the 2D profile needs to be further tuned to avoid abrupt change in Vs. Figure 12, the data quality of the real data is not good, even though it is acquired in a quiet environment. The green's function is really quite contaminated; therefore, the dispersion image is not clear. Nevertheless, understand the green's function is retrieved from the SNRS algorithm. It would be interesting to compare the green's function and dispersion image retrieved by conventional method. From there, readers would have a more explicit sense of the advantage of SNRS, if any. Figure 13, there are some differences of the results from the proposed method and the conventional active method.

Which one would be closer to real situation? A more discussion would be expected. Same comments to Figure 12 applies for Figure 16. Figure 17, is it have an figure or table to validate or compare with the 2D profile with existing wells?

---

## Referee Comment (RC2) · Anonymous Referee #2 · 27 Nov 2020

The authors present an application of passive seismic interferometry to image the subsurface of a mineral exploration area in northern Finland (down to 300 m). Passive seismic data were collected in parallel to active reflection/refraction acquisitions (during downtimes) along several linear profiles.

The main purpose of the underlying project being active seismic experiments, only a short amount of continuous passive data could be collected (hours/days). The authors try to address the challenging task of retrieving meaningful surface-wave responses from such a short duration dataset. They claim that they could achieve this despite the non-stationary and non-isotropic distribution of noise sources. For that, they used

an advanced processing algorithm called SNRS (not described in the work). They also claim (using supporting synthetic modeling) that this achievement was favored by strong local scattering conditions (local scattering helps reaching more diffuse field conditions). Using inversion of the extracted dispersion curves, they provide different 2D sections of shear-wave velocity models and propose some geological interpretations.

While I agree that the topic and goals of this work are of high interest, I do not feel at this stage that the claims made by the authors are reliably backed up in the presented work, and I think that many points should be clarified.

My major concerns are the following:

- I do not understand how the numerical simulations address the problem of non-stationary/non-isotropic noise sources. The position/angles of the sources have to be clarified, but it seems from the presented configurations that only the "pseudo-1D" case is tackled. By that I mean that the incoming noise horizontal direction seems to match the direction of the profile. This is a very favorable condition that does not address the main challenge of an off-angle dominant source of noise.

- As explained, the passive acquisition was made in parallel to active seismic acquisitions. This is a great opportunity to make detailed comparisons of active vs passive data and benchmark noise correlation/SNRS in a challenging configuration. One convincing comparison example was made for one subsurface model (Figure 13). This approach could be generalized to compare: EGFs to "active" surface-waves, dispersion curves, and other subsurface models. In my opinion, this would make a much more compelling case for the passive approach than the numerical modeling invoked above.

- The title and paper stress-out the importance of using the advanced SNRS algorithm. However, this algorithm is not described at all in the present work. Additionally, to ensure that SNRS is actually required here, a comparison with traditional Ambientnoise correlation processing could be a great addition (comparing EGFs with active data for example).

Other concerns:

- In the abstract, the passive dataset is said to contain only several hours of data. In the text, several days of acquisitions are mentioned. The exact record duration(s) should be mentioned as this is a key aspect of this work.

- The workflow from dispersion curves to subsurface models could be explained in more details.

- Why not comparing the resulting models with the results of the main project (reflection/refraction models)?

- The matching or lack of matching between model and boundary of geological units is not clearly discussed (Fig. 17).

- The quality of the figures should be improved, as well as the quality of the language.

Some detailed comments are provided in the attached annotated pdf.

Please also note the supplement to this comment:
https://se.copernicus.org/preprints/se-2020-160/se-2020-160-RC2-supplement.pdf

[Figure]

**Supplement:**

[revised manuscript text omitted]

---

## Author Response (AR1)

[revised manuscript text omitted]
. AsDue to the positions of all of the dams onin Kitinen river are well known in our experiment and all of them are located in-line with the Sakatti profile, we used the in-line position of the source in our simulationynthetic model. As an input signal for simulation, we used a real seismic signalismogram recorded by station V1 that, which was located at the shortest distance from the waterpower plant (Figure 2). The spectral-time diagram of

180 the signal is presented in Figure 7. As one can see, there are several spectral peaks with frequencies of about 5 Hz, 12.5 Hz and 20-50 Hz. According to (Antonovskaya et al., 2017; 2019), seismic noise generated by waterpower plants may correspond to a set of spectral peaks between 3.6 Hz and about 50 Hz. The oOther spectral peaks could be due to production activities (transportation, excavation etc.) at Kevitsa mine. Therefore, in this case we have complex contribution of all sources to the noise wavefield.

185 Figure 8 (a) shows synthetic seismograms of stationary wavefield produced by the signalcossesponding to, spectrogram of which is presented in Figure 7. Analysis of particle motion (Figure 8 (c)) shows that this stationary field is consisting of Rayleigh waves with apparent velocities of about 2100-2500 m/s. Figure 8(b) shows crosscorrelations of the first trace with all other traces in Fig. 8(a).  As seen, the wavefield contains also P and S waves.

Results of our synthetic modeling demonstrate that the plane wave scattered at heterogeneities satisfies condition of diffuse

190 wavefield and hence can be used to extract EGFs. The wavefield, produced by scattering of stationary signal from the waterpower plant can also be used in the cases when receivers are deployed within the first Fresnel volume area. Usage of the diffuse wavefield produced by scattering on local heterogeneities or stationary wavefield from a single source will have an advantage that in both cases the long registration time necessary for obtaining isotropic azimuthal coverage of ambient noise sources is not required. However, special analysis of the continuous data would be necessary, in order to extract the diffuse

195 wavefield from the data. For this purpose, the SNRS algorithm described earlier in Afonin et al. (2019) can be used. The technique is based on the global optimization algorithm, in which the optimized objective function is a signal-to-noise ratio of an EGF, retrieved at each iteration. Maximizing the signal-to-noise ratio of the retrieved EGF is ensured by stacking only cross- correlation functions coherent with each other and corresponding to the stationary phase area. The main idea of this

algorithm is selecting from the wavefield only those waves, which arrive from near-zero azimuths of approaches and stacking them to obtaining empirical Green's functions with a high signal-to-noise ratio. The selection is based on a global optimization algorithm, where a-priory assumptions about seismic velocities in the studied medium are used as a starting model.

[revised manuscript text omitted]
 indicates high-resolution profile; blue line indicates profile of lower-resolution.**

[Figure]

**Figure 10: The workflow for data analysis**

[Figure]

485

**Figure  11: Example of EGFs with the correspondent dispersion curve, obtained from passive seismic data for high-resolution profile shown in Fig. 9: a) EGFs; b) dispersion curve, extracted by MASW technique; c) particle motion diagrams for part of EGF indicated in (a) by red lines.**

[Figure]

490 **Figure 1112: Results of inversion of dispersion curves obtained from passive seismic data of high-resolution XSoDEx profile: a) 2D velocity model obtained by interpolation of 1D velocity models. Velocity colour scale is shown in the left. b) an example of 1D velocity model (black line) obtained by inversion of the dispersion curve in (c), corresponding to distance of 200 m in Fig. 11 12 (a); c) dispersion curve (black line), used for inversion of the velocity model in Fig. 11 12 (b). Colour scale corresponds to values of misfit function during different iteration steps in global optimization algorithm.**

[Figure]

495

**Figure 1213: Example of EGFs with correspondent dispersion curve, obtained from passive seismic data: a) EGFs on frequency band 3-8 Hz; b) dispersion curve, calculated from EGFs by MASW technique; c) particle motion diagrams of a surface wave part of an EGF indicated by red lines.**

[Figure]

500

**Figure 1314: Velocity models calculated by inversion of dispersion curves along the part of Sakatti profile marked by blue in Fig. 9. (a) S-wave velocity model obtained using passive seismic data; (b) S-wave velocity model obtained using  seismic data, which contain signals produced by controlled source.**

[Figure]

[Figure]

**Figure 1415: Crosscorrelation functions of signals, produced by Vibroseis© with the data recorded by wireless equipment at large offsets in frequency band of: a) 1-10 Hz, amplitudes normalized at maximum of each trace; b) 20-100 Hz. The signal used for cross correlation is shown in Fig. 1516.**

[Figure]

**Figure 16: An example of Vibroseis© sweep, recorded by a wireless sensor placed near the vibrator in XSoDEx experiment: a) seismogram; b) spectrogram showing the frequency content of a vibrator signal.**

[Figure]

510

**Figure 17: An example of EGFs with correspondent dispersion curve of Rayleigh wave on frequency band 5-10 Hz, obtained by advanced method of passive seismic interferometry: a) EGFs; b) dispersion curve, extracted by MASW technique; c) particle motion diagrams for surface wave part of EGF marked by red in (a).**

[Figure]

VRP – volcanic rocks, different types of relatively quartz poor eruption products mainly in Lapland;

VRR - volcanic rocks, different types of relatively quartz rich eruption products mainly in Lapland;

Q – quartzite, originally sands deposited in ocean's shore zone;

P – paraschist, originally clays deposited at the sea bottom;

GN – TTG gneiss and magmatite, plutonic rocks related to granite (3500-2750 Ma);

G – granit;

MI – layered mafic intrusions, Tornio-Koillismaa belt (2440 Ma).

- quaternary deposits, coarse-grained sorted sediments;

- fractured felsic volcanic rocks;

- fractured mafic and ultramafic volcanic rocks;

- mafic and ultramafic volcanic rocks.

515    **Figure 1718: Velocity models, calculated by inversion of dispersion curves, obtained by passive seismic interferometry for XSoDEx profiles shown in Fig. 1. The boundaries of geological units are marked by black dashed lines: a) Pomokairantie; b) Alaliesintie; c) Sakatti; d) Kuusivaarantie.**

**Author response to the interactive comment posted by Yunhuo Zhang (Referee)**

**Referee:** I have gone through the entire pre-print. It is an interesting study and useful reference. It can be considered for publication, provided some areas can be improved. Please refer below for your consideration: The title of the paper highlights the 'advanced method', which is the SNRS algorithm to estimate the green's function from diffusive ambient noise field. However, the SNRS is just referred to the author's earlier paper, without any elaboration. This makes the title not reflects the content correspondingly. Since the SNRS algorithm is already published and discussed earlier, it is suggested to amend the title accordingly, either highlighting the case study, or the attempts to characterize the noise field of the sites, etc.

**Authors:** The title of the manuscript has been changed by removing words "advanced method". The new title is "Near surface structure of Sodankylä area in Finland, obtained by passive seismic interferometry".

**Referee:** Section 4, several synthetic models are created to characterize the noise field of the site, taking into consideration of the nearby major activities. It is quite interesting and worth expanding. The assumptions of major sources need more explanation.

**Authors:** The assumption about major sources was made based on analysis of spectra, presented in the Section 3, and knowledge about locations of industrial objects, roads and other human activities. Moreover, we consider the universal source – plane wave, which is an approximation of any source located in the far field area. The corresponding text has been corrected.

**Referee:** The key message of creating the synthetic models are to support the claim that the noise field of site is diffusive. However, it is very common in elsewhere, too. Therefore, it would be better to draw some more novel conclusions from the synthetic models.

**Authors:** The main goal, except of supporting the claim that analyzed wavefield is diffuse, was also to understand how relatively high-frequency wave (dozens of Hz) may produce low-frequency (about 5-20 Hz) wavefield during scattering on heterogeneities. In our opinion, the previous synthetic modelling efforts published in literature were mainly focusing on scattering of waves from controlled sources (see, for example, Gritto et. al., 1995, Bohlen et al., 2003). In our modelling we were interested to see scattered wavefield from various types of sources, including plane wave from sources located in far field zone. In our modelling we also followed propagation of wave from these sources during time intervals that are longer than those typical for data acquisition in controlled source experiments. Spectral analysis of scattering arrivals (figure 5) shows that the plane wave with frequency of 40 Hz produces surface waves with frequencies of about 7-20 Hz during scattering. The corresponding text has been corrected.

**Referee:** Noted that each synthetic model is to simulate one type of sources. Would it better to create an overall model that combines all possible noise sources. If this one can be done, the authors may explore full waveform inversion of passive seismic waves.

**Authors:** It was shown in numerous studies (for example Wapenaar et al., 2004; Mulargia, 2012, etc.), that diffused wavefield is usually produced by the superposition of waves of numerous sources and scatterings on heterogeneities. We included these studies to the list of references. Nevertheless, our goal was to study the nature of low frequency wavefield, when all possible sources have relatively high frequencies.

**Referee:** It is essential to beef up the field acquisition in more and clearer details, e.g., the field plan out, geophone type and corner frequency, sampling rate, source signature and location for active testing, etc.

**Authors:** The chapter 2 (experiment description) has been revised and more details concerning experiment were added.

**Referee:** Figure 1 is not clear where are the blue/black lines. It is also difficult for the readers who are not familiar with Finland geology without necessary introduction.

**Authors:** The figure 1 has been changed

**Referee:** The quality of Figure 2 needs to be improved to meet the criteria of publication.

**Authors:** The figure 2 has been modified.

**Referee:** It is not clear about the caption of Figure 5 (a) that what is the distance of 2000m referring to. Figure 5(b) horizontal axis and color bar scale seem not correct, if it is a dispersion image.

**Authors:** Figure caption has been changed

**Referee:** Figure 6 (b) and 6 (c) are very interesting. It is worth expanding the explanation why these 2 directions are so different, whereby 6(b) can't see surface wave and 6 (c) can see surface wave clearly.

**Authors:** The synthetic model shows that the wave propagating from the considered source (blasts) inside the model that contains numerous irregular heterogeineities, produced only Love surface waves. In this modelling, the main objective was to clarify the absence of Rayleigh waves on seismograms, produced by blasts. The explanation of this phenomenon is not simple and might be the topic for another research and more enhanced modelling.

**Referee:** Figure 7 shows the source is mainly in 10-40 Hz, which is quite high. Please explain how such high frequency source can illuminate to 300 m below ground.

**Authors:** We show by synthetic modelling, how high-frequency waves are converted to low-frequency wavefield when scattering on heterogeneities (for example, we show that the plane wave with frequency

of 50 Hz during scattering produce wavefield with frequency of 7-20 Hz (Fig.5)). By using the SNRS algorithm, we select these scattered low-frequency waves and then analyze them.

605

**Referee:** Figure 11 (a), the 2D profile needs to be further tuned to avoid abrupt change in Vs.

**Authors:** We used only three 1D models to obtain the 2D model. We cannot smooth this model more.

610 **Referee:** Figure 12, the data quality of the real data is not good, even though it is acquired in a quiet environment. The green's function is really quite contaminated; therefore, the dispersion image is not clear. Nevertheless, understand the green's function is retrieved from the SNRS algorithm. It would be interesting to compare the green's function and dispersion image retrieved by conventional method. From there, readers would have a 615 more explicit sense of the advantage of SNRS, if any.

**Authors:** It was the topic of our previous research and one of the reasons for development SNRS algorithm. As we showed earlier (Afonin et al., 2019), conventional methods of stacking EGFs in passive seismic interferometry are not working in the seismically quiet areas of Finland where industrial activity 620 is practically absent and sources of seismic noise with high frequency (higher than 1 Hz) are rare, irregular and have low energy. In addition, high frequency noise from these sporadic sources attenuates rapidly and do not propagate to large distances. Therefore in our case, the signal and the dispersion curve presented in Figure 12, have relatively high quality. Conventional passive seismic interferometry not allowed evaluating EGF's.

625

**Referee:** Figure 13, there are some differences of the results from the proposed method and the conventional active method. Which one would be closer to real situation? A more discussion would be expected.

630 **Authors:** Figure 13 does not present results of comparison of models obtained by conventional analysis of surface waves from active sources (like MASW) with the model obtained by our method. Such comparison is not possible because vibrator does not produce Rayleigh waves of low frequencies (see Fig. 14, this is Fig. 15 in the revised manuscript). Figure 13 (Figure 14 in the revised manuscript) shows comparison of velocity models, obtained by analysis of wavefield produced by scattering of signal of 635 vibrator (Fig. 13 (b)) with the wavefield produced by scattering of waves of unknown source (Fig. 13 (a)). That is why in our study we compared them not to each other, but to a priori geological information. Both of them do not contradict the geological information. From Figure 12 (b) one can see that the width of error bars of dispersion curves are about 500 m/s. Differences in velocities between two 2D models (Figure 13) are within these limits.

640

**Referee:** Same comments to Figure 12 applies for Figure 16. Figure 17, is it have an figure or table to validate or compare with the 2D profile with existing wells?

**Authors:** Unfortunately, there are too few data from wells in the studied area available for comparison. The wells drilled by Geological Survey of Finland does not penetrate deep, and the data of wells drilled by exploration companies are not available for research organisations. We compared our results with geological information (boundaries of geological units) and with the data about rock properties and composition in XSoDEx area summarized by Leväniemi et al. (2018). In this study no direct measurements of S-wave velocities were made. The presented results are the first detailed information about shear wave seismic velocities in the subsurface for the studied area.

**Refernces**

Afonin, N., Kozlovskaya, E., Nevalainen, J., Narkilahti, J.: Improving the quality of empirical Green's functions, obtained by cross-correlation of high-frequency ambient seismic noise, Solid Earth, 10(5), 1621-1634, https://doi.org/10.5194/se-10-1621-2019, 2019

Bohlen, T., Mueller, Ch. And Milkereit, B.: Elastic Seismic Wave Scattering from Massive Sulfide Orebodies. On the role of Composition and Shape. In: Eaton, D.W., Milkereit, B., and Salisbury, M. Hardrock Seismic Exploration. Geophysical Developments No. 10, Society of Exploration Geophysics, pp. 70-89, 2003.

Gritto, R., Korneev, V. A., & Johnson, L. R. (1995). Low-frequency elastic-wave scattering by an inclusion: limits of applications. Geophysical Journal International, 120(3), 677-692.

Leväniemi, H., Melamies, M., Mertanen, S., Heinonen, S., Karinen, T.: Petrophysical measurements to support interpretation of geophysical data in Sodankylä, northern Finland, Geological Survey of Finland Open File Work Report 25/2018, 2018

Mulargia, F. (2012). The seismic noise wavefield is not diffuse. The Journal of the Acoustical Society of America, 131(4), 2853-2858.

Wapenaar, K.: Retrieving the Elastodynamic Green's Function of an Arbitrary Inhomogeneous Medium by Cross Correlation, Physical review letters, 93(25), 254301, https://link.aps.org/doi/10.1103/PhysRevLett.93.254301, 2004

**Author response to the interactive comment posted by Anonymous referee #2**

First of all, we would like to point out that we use the old figure numbers in our replies. In the revised manuscript one new figure was added and numeration of figures has changed.

**Referee:** The authors present an application of passive seismic interferometry to image the subsurface of a mineral exploration area in northern Finland (down to 300 m). Passive seismic data were collected in parallel to active reflection/refraction acquisitions (during downtimes) along several linear profiles. The main purpose of the underlying project being active seismic experiments, only a short amount of continuous passive data could be collected (hours/days). The authors try to address the challenging task of retrieving meaningful surface-wave responses from such a short duration dataset. They claim that they could achieve this despite the non-stationary and non-isotropic distribution of noise sources. For that, they used an advanced processing algorithm called SNRS (not described in the work). They also claim (using supporting synthetic modeling) that this achievement was favored by strong local scattering conditions (local scattering helps reaching more diffuse field conditions). Using inversion of the extracted dispersion curves, they provide different 2D sections of shear-wave velocity models and propose some geological interpretations. While I agree that the topic and goals of this work are of high interest, I do not feel at this stage that the claims made by the authors are reliably backed up in the presented work, and I think that many points should be clarified.
My major concerns are the following:
- I do not understand how the numerical simulations address the problem of nonstationary/ non-isotropic noise sources. The position/angles of the sources have to be clarified, but it seems from the presented configurations that only the "pseudo-1D" case is tackled. By that I mean that the incoming noise horizontal direction seems to match the direction of the profile. This is a very favorable condition that does not address the main challenge of an off-angle dominant source of noise.

**Authors:** We considered different positions and types of sources (in line with the profile in surface (vibrator, waterpower plant dam and blast), sources out-of-line and also plane wave arriving with different incidence angles). Such positions were selected because in our study area we know positions of vibrator, dam and the active mine. In simulations we were just trying to represent situation with these real noise sources we identify in our experiment areas. In other experiments the sources may have different positions with respect to profiles, of course, but this is out of scope of our case-study paper. Moreover, when we considered a plane wave, which is an approximation of any source in far field area, we tested several incidence angles and azimuths of arrivals. In all cases, we noticed scattered arrivals with characteristics (polarization, dispersion) of Rayleigh waves and the similar apparent velocities for all considered cases. The similar scattered arrivals we noticed in synthetic seismograms, produced by vibrator, explosive source or dam located in-line with the profile. In the cases when the source was a plane wave or vibrator, we noticed that scattered arrivals have lower frequencies than the source, and that the scattered waves have polarization and apparent velocities typical for Rayleigh waves. Therefore, our modelling suggests that seismic waves from considered sources are converted to diffused wavefield of lower frequencies when scattering. Our modelling is in line with results of previous theoretical studies (for example, Gritto

et. al., 1995; Wapenaar, 2004). The analysis of real data recorded in XSoDEx project and presented in our papers proves our suggestions.

**Referee:** As explained, the passive acquisition was made in parallel to active seismic acquisitions. This is a great opportunity to make detailed comparisons of active vs passive data and benchmark noise correlation/SNRS in a challenging configuration. One convincing comparison example was made for one subsurface model (Figure 13). This approach could be generalized to compare: EGFs to "active" surface-waves, dispersion curves, and other subsurface models. In my opinion, this would make a much more compelling case for the passive approach than the numerical modeling invoked above.

**Authors:**
Direct comparison of the results from the proposed method and from the conventional methods based on surface wave analysis from active sources (like MASW, for example) is impossible because vibrator does not produce Rayleigh waves of low frequencies that we used in our method (Fig. 14). We pointed out on this in the text of Section 3 and this was the main motivation for our synthetic modelling. Figure 13 shows comparison of velocity models obtained by analysis of wavefield produced by scattering of signal from vibrator (Fig. 13 (b)) with the model obtained using scattering of waves of unknown source (Fig. 13 (a)). Therefore, in both cases, the scattering field is used and we applied our method in both cases in order to obtain EGFs. That is why neither of the models can be considered as a benchmark. We compare both models mainly to geological information and we find no contradiction. From Figure 12 (b) one can see that the width of error bars of dispersion curves are about 500 m/s. Differences in velocities between two 2D models (Figure 13) are within these limits.

**Referee:** The title and paper stress-out the importance of using the advanced SNRS algorithm. However, this algorithm is not described at all in the present work. Additionally, to ensure that SNRS is actually required here, a comparison with traditional ambient noise correlation processing could be a great addition (comparing EGFs with active data for example).

**Authors:** It was the topic of our previous research and one of the reasons for development SNRS algorithm. The detailed description of the algorithm and results of its testing with real data in two different areas with different type of ambient noise sources are already published in the paper by Afonin et al. (2019). As we showed in this previous study, using of conventional passive seismic interferometry for extracting empirical Green's functions from seismic noise of high frequency (higher than 1 Hz) is practically impossible in seismically quiet areas of Finland where significant industrial activity is absent and sources of seismic noise with high frequency are rare, irregular and have low energy. In addition, high frequency noise from these sporadic sources attenuates rapidly and do not propagate to large distances, so simple noise crosscorrelation does not work.
Concerning comparison with the active data, see our reply to the question above.

**Referee:** In the abstract, the passive dataset is said to contain only several hours of data. In the text, several days of acquisitions are mentioned. The exact record duration(s) should be mentioned as this is a key aspect of this work.

**Authors:** We used data intervals with durations varying from several hours to couple of days (it was the period when vibrator was in reparation). When we analyzed parts of records with vibrosource in Section 6 it was one day of continuous data. When we analyzed passive seismic data without vibrator, it was a couple of days. Nevertheless, we analyzed continuous seismic data without selecting parts of records with signal of vibrator. The SNRS method is selecting automatically parts of the record for retrieving EGFs.

**Referee:** The workflow from dispersion curves to subsurface models could be explained in more details.

**Authors:** The main steps of data processing are described in Section 5. They are well known (MASW for obtaining dispersion curves, inversion of dispersion curves using Geopsy). References and a new figure (Figure 10) describing the workflow is added to the text.

**Referee:** Why not comparing the resulting models with the results of the main project (reflection/ refraction models)?

**Authors:** To the moment, detailed models from analysis of reflected and refracted P-waves are still not published in regular papers. That is why we decided not to include any preliminary results of these studies into our paper that concentrates mainly on analysis of S-wave velocities down to 300 m. We compared our results mainly with petrophysical and geological information.

**Referee:** The matching or lack of matching between model and boundary of geological units is not clearly discussed (Fig. 17).

**Authors:** The XSoDEx study is the first one, where upper subsurface of this area was studied in such details. Our study reveals boundaries between major lithological units in the area. However, it is necessary to take into account that the boundaries of geological units in this area are known with certain precision, depending on density of points for geological sampling. That is why certain discrepancy between previously defined boundaries of geological units and geophysical information is possible. Geologists need input from geophysical studies in order to upgrade their knowledge.

**Referee:** The quality of the figures should be improved, as well as the quality of the language.

**Authors:** The quality of the figures has been improved and we made additional work on language.

**Referee:** Some detailed comments are provided in the attached annotated pdf. Please also note the supplement to this comment: https://se.copernicus.org/preprints/se-2020-160/se-2020-160-RC2-supplement.pdf

**Authors:** Replies to these comment provided in attached.

---

## Referee Report (RR1)

**Review of the first revision of "Near surface structure of Sodankylä area in Finland, obtained by passive seismic interferometry" by Afonin et al.**

During this first round of revision, the authors partially replied to the comments/suggestions made by the reviewers, and improved the quality of their manuscript. However, one of my main concerns remains, namely the clarity and the relevance of the numerical simulation section.

Following my previous comments, the authors explained that they simulated an incoming plane-wave with an azimuthal angle of 40 degrees relative to the profile. Having an off-angle source is indeed crucial to prove that the surface-wave reconstruction could in theory work in their real-case scenario. However, looking at the geometry of the simulation model, namely the absence of heterogeneity along the Y-axis, it is puzzling to me to understand how a surface-wave with correct velocity could be extracted from this configuration.

Another puzzling observation is the presence of Love waves (polarized along the Y-axis) generated by the explosive source located along the profile/X-axis. Again, for reasons of symmetry of the model, I do not understand how this is possible. Is it a misunderstanding about the model configuration, maybe unmentioned heterogeneity? I once again suggest to clarify the figure regarding the simulation model and source configurations.

A third point of concern on the simulation topic relates to an addition/clarification that the authors have made in this new version and in their response to reviewer 1. They claim that "diffuse wavefield consisting of low-frequency (5-20 Hz) surface waves (Rayleigh) can be produced by scattering of a high-frequency (50 Hz in our case) plane wave at velocity heterogeneities." They stated that "the main goal, except of supporting the claim that analyzed wavefield is diffuse, was also to understand how relatively high-frequency wave (dozens of Hz) may produce low-frequency (about 5-20 Hz) wavefield during scattering on heterogeneities". If I am not mistaken, solving the standard elastic-wave equation does not allow such non-linear conversions to occur (did the authors simulate non-linearities?). The only possibility is that non-dominant frequencies already present in the source spectrum can be "selected" by scattering. Because there is virtually no noise in simulations, any non-zero frequency content of the source could be potentially revealed by appropriate scatterers.

While I acknowledge that the study presents a strong interest if absent of artifacts, It is still difficult for me to recommend the paper for publication in its current state because of the mentioned interrogations. I would suggest to ask for the additional opinion of someone with expertise in such numerical simulations and in non-standard Green's function extraction from ambient noise.

Best regards

---

## Referee Report (RR2)

Review report on « Near surface structure of Sodankylä area in Finland, obtained by passive seismic interferometry " by Nikita Afonin et. al.

The deduced results are interesting and can be considered for publication. provided some areas can be improved. The authors applied seismic interferometry to passive seismic data to retrieving surface-wave and to image the near subsurface structures related to mineral exploration. They used SNRS as part of the processing procedure to retrieve the green function. However, it is not clear to me how and why they used this algorithm. They also performed synthetic tests which the goal is unclear to me. Although the resulted 2D sections of shear-wave velocity models from inverting of the dispersion curves and the geological interpretations are very interesting, there are some points that need consideration.

My major concerns are the following:

1- In general the paper has a good structure and is well written but there are some (unnecessary) statements in the abstract, introduction, and also other sections that need to be rephrased on removed to improve the manuscript (ms). Also, there are some typos that need to be checked.

2- Lines 116-117: I can't see how they concluded about the distribution of the sources from PSD of the signals in Fig. 2. Usually, for different frequency band that will be used in a study, we apply a beamforming or FK analysis to locate the main source but I don't see such an analysis and there are just some statement in the text that is not enough in my opinion.

3- Line 170: I can't see any spectral peaks at those frequencies in Fig. 2. I think the figures cross-referencing in the whole ms need some improvement as there are some sentences without (correct) cross-referencing to the corresponding figure.

4- Line 178-187: I do not understand how the numerical simulations results address any of the problems and how it helped the authors as the results demonstrations in Figs 6 and 8 is different. They didn't show the cross-correlation results for the single source. And to me, it's not clear how this helped the authors as they didn't apply their processing technique to the simulated data. And there is no analysis of how applying SNRS improves the results. I understand it has been explained in another earlier paper but it would be useful, if possible, to do the comparison for this simulated data as well.

5- Line 200: In the processing, you mentioned 1-100Hz bandpass filtering but in all the figures for the dispersion analysis you only used frequencies <50, is there any specific reason to use 1-100 and not 50Hz for filtering? And in lines 245, what did you use to eliminate the surface-wave? Is it a notch filter?

6- Line 263: Where are the drilling locations? I couldn't find any map about their location.

7- Lines 268-270: I don't understand the meaning of this paragraph, usually the non-stationary phases increase the apparent velocity so there is no need for this statement.

8- Line 287: Again, I can't see how you concluded if the noise sources are isotropic or not?

9- Could you please explain why you used different packages/codes for dispersion curve calculation for different parts of the data? Somewhere you used "Geopsy" and then changed to "MASW"? I know they both do the job but maybe it would be better to be consistent if you want to compare the results.

10- General suggestion on figures: Maybe you can combine Figures 6 and 8 for a better comparison. There is no need to plot the particle motions. Use a scalebar for some of the figures is possible.

Hope these questions and comments help in the improvement of ms.
Sincerely
M. Rezaeifar

---

## Author Response (AR2)

**Replies to Anonymous Referee #2**

**Referee:**
Review of the first revision of "Near surface structure of Sodankylä area in Finland, obtained by passive seismic interferometry" by Afonin et al.

During this first round of revision, the authors partially replied to the comments/suggestions made by the reviewers, and improved the quality of their manuscript. However, one of my main concerns remains, namely the clarity and the relevance of the numerical simulation section.
Following my previous comments, the authors explained that they simulated an incoming plane-wave with an azimuthal angle of 40 degrees relative to the profile. Having an off-angle source is indeed crucial to prove that the surface-wave reconstruction could in theory work in their real-case scenario. However, looking at the geometry of the simulation model, namely the absence of heterogeneity along the Y-axis, it is puzzling to me to understand how a surface-wave with correct velocity could be extracted from this configuration.

**Authors:** The passive data used in our paper were measured along XSodEx 2-D reflection profiles. According to the conventional practice for planning of 2-D seismic profiles, their location is across the major geological units in the area, but not along these units (Figure 1). That is why we considered this particular case in our numerical modelling, namely, the case when the data is acquired along profile crossing geological units. The purpose of our study was to interpret the data of this particular experiment and the numerical modelling of all the possible structures and situations was not the main purpose of our study. However, we made our own numerical modelling for this particular case because in previous studies that use numerical modelling (for example, Bohlen et al., 2003) the authors usually model propagation during short times comparable to registration times used in active experiments. In our modelling we were mainly interested to consider the wavefield, produced by multiple scattering of a plane wave on heterogeneities (not the waves itself, produced by source). For this, it was necessary to consider longer times (more than two seconds, see, for example, Figure 4).
As was shown in numerous theoretical studies (references are given in the manuscript), the waves, produced by scattering of a plane wave on heterogeneity, are scattered in all directions. In that case, each heterogeneity is a source of the scattered waves, which propagate in all directions with true velocity (velocity that corresponds to elastic properties of the medium). If there are many heterogeneities, the scattered waves form the resulting wavefield that may be considered in a diffused field approximation (Shapiro and Campillo, 2004). We think that our numerical modelling proves results of theoretical studies one more time, showing that such wavefield can exist in real situations, because the model of the medium and position of sources were selected to be as close as possible to the real experiment configuration.

**Referee:** Another puzzling observation is the presence of Love waves (polarized along the Y-axis) generated by the explosive source located along the profile/X-axis. Again, for reasons of symmetry of the model, I do not understand how this is possible. Is it a misunderstanding about the model configuration, maybe unmentioned heterogeneity? I once again suggest to clarify the figure regarding the simulation model and source configurations.

**Authors:** We added particle motion diagram to the Figure 6. The polarization shows that considered wave is Love wave. Thin upper layer with low velocities that models quaternary sediments most probably causes presence of the Love waves.

**Referee:** A third point of concern on the simulation topic relates to an addition/clarification that the authors have made in this new version and in their response to reviewer 1. They claim that "diffuse wavefield consisting of low-frequency (5-20 Hz) surface waves (Rayleigh) can be produced by scattering of a high-frequency (50 Hz in our case) plane wave at velocity heterogeneities." They stated that "the main goal, except of supporting the claim that analyzed wavefield is diffuse, was also to understand how relatively high-frequency wave (dozens of Hz) may produce low-frequency (about 5-20 Hz) wavefield during scattering on heterogeneities". If I am not mistaken, solving the standard elastic-wave equation does not allow such non-linear conversions to occur (did the authors simulate non-linearities?). The only possibility is that non-dominant frequencies already present in the source spectrum can be "selected" by scattering. Because there is virtually no noise in simulations, any non-zero frequency content of the source could be potentially revealed by appropriate scatterers.

**Authors:** We used numerical simulation to see how the waves produced by possible sources existing in our study area can be scattered at the velocity heterogeneities corresponding to real geological structures. General studies of the scattering phenomena was not a goal of the current work. We agree that some of the numerical simulations (for example, Ryberg et al., 2000) suggest the selective properties of scattering on heterogeneities. However, the presence of non-dominant frequencies was not possible in the case of plane wave that we modelled. In our numerical modelling, the plane wave was produced by multiple sources with source time functions shaped as delta functions. Therefore, presence of any additional harmonics is not possible in our numerical modelling example.

References:
Shapiro, N. M., & Campillo, M. (2004). Emergence of broadband Rayleigh waves from correlations of the ambient seismic noise. Geophysical Research Letters, 31(7).
Bohlen, T., Müller, C., Milkereit, B., Eaton, D. W., & Salisbury, M. H. (2003). Elastic seismic wave scattering from massive sulfide orebodies: on the role of composition and shape. Hardrock seismic exploration: SEG, 70, 89.
Ryberg, M. Tittgemeyer, F. Wenzel, 2000. *Geophysical Journal International*, 141, 3, 787–800, https://doi.org/10.1046/j.1365-246x.2000.00117.x

**Replies to Referee #3: Rezaeifar Meysam**

**Referee:**
Review report on « Near surface structure of Sodankylä area in Finland, obtained by passive seismic interferometry " by Nikita Afonin et. al.
The deduced results are interesting and can be considered for publication. provided some areas can be improved. The authors applied seismic interferometry to passive seismic data to retrieving surface-wave and to image the near subsurface structures related to mineral exploration. They used SNRS as part of the processing procedure to retrieve the green function. However, it is not clear to me how and why they used this algorithm.

**Authors:** We used this algorithm because the length of passive seismic records was not enough for using conventional passive seismic interferometry. Although some authors used shorter lengths of records to retrieve EGF's (e.g. Draganov et. al., 2007) in our case it was not possible, probably due to specific ambient noise features. The studied area is relatively seismically quiet and the amplitudes of ambient seismic noise in the high frequency range are low. The SNRS algorithm allows obtaining EGFs by selecting crosscorrelation functions, which corresponds to parts of wavefield, produced by scattering of waves from the strongest sources. This is achieved by the global optimization algorithm. Further stacking of selected cross-correlation functions allows to increase significantly the quality of retrieved EGFs. The algorithm itself as well as its advantages and possibilities is described in detail

in our previous work (Afonin et. al., 2019). We also describe the algorithm in Supplementary Material. In our revised manuscript we also show comparison of EGFs retrieved using the traditional stacking of crosscorrelation functions and EGFs obtained using SNRS algorithm (Figure 17)

**Referee:** They also performed synthetic tests which the goal is unclear to me.

**Authors:** The main reason for the modelling was checking the possibility of using passive seismic interferometry in our experiment. As the main condition for this is existing of diffuse wavefield, the purpose of our modelling was to demonstrate how this diffuse wavefield is produced. In our numerical simulation, we used configuration of profiles, structural features of the studied medium, positions and characteristics of dominant noise sources as close as possible to the real situation in the XSodEx experiment. We show that scattering of a plane wave on heterogeneities produces scattered wavefield of relatively low frequency. If there are multiple heterogeneities, this wavefield can be considered as diffuse field approximation (Shapiro and Campillo, 2004). Therefore, using passive seismic interferometry is justified, because the main condition for its application is satisfied.

**Referee:** Although the resulted 2D sections of shear wave velocity models from inverting of the dispersion curves and the geological interpretations are very interesting, there are some points that need consideration.
My major concerns are the following:
1- In general the paper has a good structure and is well written but there are some (unnecessary) statements in the abstract, introduction, and also other sections that need to be rephrased on removed to improve the manuscript (ms).
Also, there are some typos that need to be checked.
2- Lines 116-117: I can't see how they concluded about the distribution of the sources from PSD of the signals in Fig. 2. Usually, for different frequency band that will be used in a study, we apply a beamforming or FK analysis to locate the main source but I don't see such an analysis and there are just some statement in the text that is not enough in my opinion.

**Authors:** We could not use FK or beamforming because our sensors were installed along semi-2D profiles. However, our research area is seismically very quiet and the seismic noise of considered frequencies (higher than about 2-5 Hz) is mainly produced by known sources of human activity (transport, active mines and other industrial objects) or natural sources such as rivers. From the map (Fig. 2), one can see that such objects are not distributed homogeneously around the XSoDEx study area. Therefore, the main condition of using conventional passive seismic interferometry (the azimuthal distribution of noise sources is homogeneous) is not satisfied. That is why we consider one more possible source type that is scattering of plane waves. Our modelling shows that plane waves scattered at heterogeneities are producing the resulting wavefield that can be considered as diffuse field approximation. Nevertheless, these scattered waves are weak, that is why we used SNRS algorithm for retrieving EGFs.

**Referee:** 3- Line 170: I can't see any spectral peaks at those frequencies in Fig. 2. I think the figures cross-referencing in the whole ms need some improvement as there are some sentences without (correct) cross-referencing to the corresponding figure.

**Authors:** Line 170 described result of spectral-time analysis, presented in figure 7: "The spectral-time diagram of the signal is presented in Figure 7. As one can see, there are several ranges of frequencies with some increasing of amplitudes (about 5 Hz, 12.5 Hz and 20-50 Hz)." We corrected the sentence to avoid misunderstanding (word "peaks" was changed to "frequency ranges").

**Referee:** 4- Line 178-187: I do not understand how the numerical simulations results address any of the problems and how it helped the authors as the results demonstrations in Figs 6 and 8 is different. They didn't show the crosscorrelation results for the single source. And to me, it's not clear how this helped the authors as they didn't apply their processing technique to the simulated data. And there is no analysis of how applying SNRS improves the results. I understand it has been explained in another earlier paper but it would be useful, if possible, to do the comparison for this simulated data as well.

**Authors:** We show crosscorrelation functions estimated from synthetic data in Figure 8. Lines 178-187 describes results, presented in figures 4 and 8. We show that plane waves scattered on heterogeneities may be considered as diffused wavefield approximation and hence can be used for retrieving EGFs. We show in Figure 4, that scattered waves have velocities and polarizations of Rayleigh waves. In the case of using a stationary wavefield originating from the dam, we can use this signal itself.

We also need to satisfy the condition that the source is in-line with the profile inside the first Fresnel volume (Wapenaar, 2010). In some cases (for example, Sakatti line in Figures 1 and 2), the dam is located in-line with the profile and we can use superposition of both the sources (scattered wavefield, produced by scattering of plane wave and the wavefield produced by the dam). Nevertheless, for Pomokairantie profile, for which only a scattered wavefield of a plane wave exists, we can also use diffused field approximation in order to estimate EGFs. As this wavefield is too weak, we have to use the SNRS algorithm.

**Referee:** 5- Line 200: In the processing, you mentioned 1-100Hz bandpass filtering but in all the figures for the dispersion analysis you only used frequencies <50, is there any specific reason to use 1-100 and not 50Hz for filtering?

**Authors:** This is a part of widely used "standard" procedure of data processing (Figure 10), in which bandpass filtering is used at pre-processing stage. We used this frequency band because we tried to retrieve the body wave parts of EGFs. The results for body waves not satisfied us hence we decided not to include them in the manuscript. However, the description of the procedure of data preparation for EGFs calculation remains the same.

**Referee:** And in lines 245, what did you use to eliminate the surface-wave? Is it a notch filter?

**Authors:** We did not eliminate surface waves. Line 245 describes the processing of the reflection experiment data presented in Buske et al. (2019). We just show that vibrator produced the seismic signal with frequencies no lower than about 10 Hz (Figure 16). At the same time, we retrieved surface waves which frequencies of about 2-3 Hz from our passive data. We just demonstrate (also by numerical simulation) that we analyzed not signal of vibrator itself, but scattered wavefield. This wavefield is result of scattering of waves, produced by vibrator.

**Referee:** 6- Line 263: Where are the drilling locations? I couldn't find any map about their location.

**Authors:** Drilling results along all XSodEx profiles are summarized in Master Thesis by Karjalainen (2019), which can be found in open access (http://jultika.oulu.fi/Search/Results?lookfor=Karjalainen+Jari). This thesis is using a lot of information about previous geological studies. We used the results summarized in this thesis, not drilling data itself. The reference is added to the list of references.

**Referee:** 7- Lines 268-270: I don't understand the meaning of this paragraph, usually the non-stationary phases increase the apparent velocity so there is no need for this statement.

**Authors:** The paragraph has been rephrased.

**Referee:** 8- Line 287: Again, I can't see how you concluded if the noise sources are isotropic or not?

**Authors:** As we could not use FK or beamforming due to the linear configuration of sensor deployments, we suggested that the main noise sources in the frequency range of interest are related to human activity (roads, industrial objects like mines, etc.) or natural objects (rivers). The study area is not densely occupied and location of all such sources is well known. Only a few such objects are known in the studied area and the noise from them is coming from several azimuths only. In other words, we were dealing with non-isotropic azimuthal distribution of noise sources and, as a result, it was not possible to apply the conventional method of passive seismic interferometry that requires homogeneous azimuthal distribution of noise sources.

**Referee:** 9- Could you please explain why you used different packages/codes for dispersion curve calculation for different parts of the data? Somewhere you used "Geopsy" and then changed to "MASW"? I know they both do the job but maybe it would be better to be consistent if you want to compare the results.

**Authors:** We used the same software for all parts of data ("Geopsy"). "MASW" is mentioned as a method, but we used the Geopsy software for extraction of dispersion curves that were used for MASW.

**Referee:** 10- General suggestion on figures: Maybe you can combine Figures 6 and 8 for a better comparison. There is no need to plot the particle motions. Use a scalebar for some of the figures is possible.
Hope these questions and comments help in the improvement of ms.
Sincerely
M. Rezaeifar

**Authors:** We were trying to improve the quality of all figures, taking into account also comments of other reviewers. We cannot combine Figs 6 and 8 because in Fig. 8 we show crosscorrelation estimated from synthetic data. The particle motion analysis was used to analyse polarization and type of the wave.

We are very thankful for the comments that helped us to improve our manuscript.

References:

Draganov, D., Wapenaar, K., Mulder, W., Singer, J., & Verdel, A. (2007). Retrieval of reflections from seismic background-noise measurements. Geophysical Research Letters, 34(4).

Shapiro, N. M., & Campillo, M. (2004). Emergence of broadband Rayleigh waves from correlations of the ambient seismic noise. Geophysical Research Letters, 31(7).

Wapenaar, K., Draganov, D., Snieder, R., Campman, X., & Verdel, A. (2010). Tutorial on seismic interferometry: Part 1—Basic principles and applications. Geophysics, 75(5), 75A195-75A209.

Karjalainen, J.: Ambient noise H/V spectral ratio and its application for estimating thickness of overburden in XSoDEx project (Master's thesis), Oulu, Oulun yliopisto, teknillinen tiedekunta, kaivannaisala, geologia, 181 pp, 2019

**Replies to Anonymous Referee #4**

**Referee:** Afonin et al conduct seismic interferometry on seismic data in Finland which was originally acquired for active source purposes. They produce velocity models of the shallow (<300m) surface and do numerical modelling of likely seismic sources in the area. The work is important to show how data acquired in non-optimal geometry and with a short recording time can be used for interferometry. However substantial details are missing that limit the conclusions that the authors draw - particularly details on the acquisition and modelling. The authors also do not compare their velocity models to those acquired with the active source, which should be a key part of their analysis. The text and figures also need improvements (particularly maps and poor colour scales).

**Authors:** We were trying to improve the figures as recommended. Concerning comparison of our model with the models obtained by active source, such direct comparison is practically not possible. First, we used surface waves and obtained S-wave velocity model, while controlled-source reflection experiment was not aiming to study S-wave velocities, but near-vertically reflected P-waves. Secondly, in our study we investigated different depths range. The depths of investigation for reflection survey was about 3 km (Buske et al., 2019) and we studied the structure from several meters down to about 300 m. Thirdly, it was not possible to apply classical MASW method for analysis of surface waves produced by controlled source because their frequencies are higher than 10 Hz and they are not penetrating to the same depth as the surface waves used in our study. This is shown in Fig. 16 and discussed also in the text. That is why we cannot compare MASW results obtained by controlled source and by passive seismic methods.

**Referee:** Regarding the conclusions of the work - I agree that high velocity contrasts are needed to produce scattering but the authors use only 1 subsurface model so I don't think they can claim they show this in their numerical modelling (they could also add reference to other modelling work).

**Authors:** In conclusions, we refer to theoretical studies by Wapenaar (2004) and Wapenaar and Thorbecke (2013). They considered the phenomenon of origin of diffuse wavefield theoretically. In our paper, we made the numerical simulation for the model that is closest to the real geological structure in our study area, using the sources and receivers geometry that existed in our particular area during our particular experiment. We found out that our results agree with these theoretical studies. The goal was to explain the wavefield only in our particular case and test whether the passive seismic interferometry can be applied to the data acquired in the XSodEx experiment (study of near surface structure of the Sodankylä area). We added references to others modelling works into the text.

**Referee:** The other conclusion that the SNRS method is needed to produce EGFs is also not shown – can EGFs also be produced without this algorithm?

**Authors:** In our case, conventional method of passive seismic interferometry not working because of relatively short length of seismic record. In our previous work (Afonin et. al., 2019), we showed that using SNRS algorithm allows to decrease the necessary length of passive seismic record. We added the EGFs evaluated without using SNRS algorithm to Figure 17, for comparison. As one can see, there is no coherent waveforms that can be used for evaluation of dispersion curves.

**Referee:** Major comments

Acquisition – the description of the acquisition set up is confusing and needs more details, a more detailed figure would help to clarify it. Did all 4 lines have the same set up? Were the profiles gathered along roads or footpaths? How many geophones / nodes were active at once and for how long / what was their max offset? Did the authors use the geophones that were deployed in the reflection survey (surely they were not active for very long)? We need more information on the recording time – what was the average recording time between station pairs, and the lowest and highest? Do the authors remove time periods when there are active sources?

**Authors:** The part of the text about description of acquisition has been enhanced and modified.

**Referee:** Numerical modelling – From the modelling, the authors claim that surface waves are produced that can be used to derive EGFs. But what it is about their model that produces surface waves (e.g. from plane waves arriving from depth)? This could be more impactful if the authors used different velocity models and described how the presence of surface waves changed. E.g what is the effect of the shallow velocity layer and sharp velocity contrasts that are in the model?

**Authors:** We used the velocity and structural model of the medium, as well as position of possible sources, which are as close as possible to the situation in our real experiment. The model represents a typical for northern Finland situation when the old (Precambrian) weathered bedrock with felsic-to-mafic lithologies is overlaid by thin quaternary sediments formed after last glaciation. It was necessary for checking the possibility of using passive seismic interferometry in diffused field approximation in our concrete applied problem: interpretation of passive data of the XSodEx experiment. We did not have an ambitious goal to model all the possible cases where scatteres can be present. This is an interesting subject, but it is better to address in a separate paper.

**Referee:** The authors should also try to describe why Rayleigh waves are not produced in some cases.

**Authors:** In our modelling, we considered the case when seismic sensors are installed along the profile across the high velocity heterogeneity. Similar cases were considered, for example, by Ikeda and Tsuji, 2016. They showed that in some cases Rayleigh waves could be absent.

**Referee:** What is also the effect of moving the source location from parallel to perpendicular to the line?

**Authors:** In our study the main source type considered was plane wave. Of course, we consider several incidence angles for plane wave in our modelling, but the results (scattered wavefield) were the same. That is why we decided not to include all of them in our paper because that is a case-study aiming to interpret real data. Concerning other sources, we used their location closest to location of real sources in our area. We know positions of the main sources (the Kevitsa mine and its open pit is a source of both continuous noise and blasts, the dam is located in-line with some profiles and it is the source of continuous noise). Quite naturally, the vibrosource was also moved along the profiles in reflection experiment.

**Referee:** The authors should also place their modelling work in the context of previous work and compare it.

**Authors:** The references to the previous numerical modelling results are added to our manuscript. However, we would like to point out that in these previous studies the authors were trying to model

the wavefield from active sources and explain the wavefield recorded during short registration times typical for controlled-source experiments. In our case we considered longer times, necessary to study the propagation of the scattered wavefield in passive seismic experiments. Our aim was to explain behavior of the wavefiled in our specific case study.

**Referee:** I also question whether modelling the presence of directional Rayleigh waves is enough to conclude that the wavefield is diffuse (e.g. L178).

**Authors:** It is theoretically shown in such studies as Wapenaar (2004) and Wapenaar and Thorbecke (2013) that scattered wavefield may be considered in diffused field approximation. Scattered wavefield is also widely used in coda wave interferometry (e.g. Camplillo and Paul, 2003; Snieder et. al., 2002; Snieder, 2006; etc.). In fact, in our work we modelled coda waves originating from some sources typical for our study area. Therefore, we can consider it in diffused field approximation.

**Referee:** SNRS method – more details should be added about this method not just giving the reference. From what I understand, the method works by stacking green's functions but only those that improve the signal to noise ratio. I am really concerned that this will not result in the 'true' green's function but only that with the highest 'signal' by whatever way signal is measured.

**Authors:** We cannot include the full description of the method into our manuscript, as it is already published in the other paper (Afonin et. al., 2019). More detailed description is added to Supplementary Material.

**Referee:** Figures could be substantially improved:
Figure 1 – please add latitude, longitude. The aerial map is too low quality, I cannot see it clearly. In addition to the geological map, please add a similar sized topographical map showing roads, towns etc.

**Authors:** Figure 1 has been improved.

**Referee:** Figure 2 – the background map is too dark and low resolution. Please add a scale and north. Caption: four stations or 6?

**Authors:** Figure 2 has been improved. Figure caption has been corrected.

**Referee:** Fig 9 need scale, is the view looking verticall down? Annotate virtual source locations.

**Authors:** Figure 9 has been changed

**Referee:** The colour scales for velocity models in Fig 12, 14 , 18 are poor. They are not continuous and have been set in some way to make it look like blocks. It is difficult to assess the 2D velocity models with such colour scales. The colour scales are not the same between the models presented. Add locations of 1D velocity profiles to the 2D velocity lines.

**Authors:** The colour scales were selected to better separate velocities corresponding to different rock types (in particular, quaternary sediments from bedrock). Different colour scales also were selected for improving visualization of results and simplify the comparison of velocity models. The 2D model looks like blocks, because they were interpolated from limited number of 1D models. We did not apply smoothing, as it may have negative impact on the interpretation of the results. Positions of 1D models are already marked by ticks and captions at every 500 m of profile.

**Referee:** The writing could be improved – I have listed some cases below but there are many more examples.

**Authors:** The language using has been improved.

**Referee:** Other comments
P2L34 'actual task', please rephrase (important task?)

**Authors:** Done

**Referee:** P2L36 remove 'than earlier'

**Authors:** Done

**Referee:** L49 remove 'possibility'

**Authors:** In L49 there is no word 'possibility'

**Referee:** L59 new paragraph

**Authors:** Done

**Referee:** L66 I don't know what the authors mean by 'directional scatterer'?

**Authors:** A detailed description of directional scatterrers was presented in the paper (Wapenaar and Thorbecke, 2013). We provided a reference in the text.

**Referee:** L77 the authors should describe why this is an 'advanced method'.

**Authors:** The sentence has been rephrased.

**Referee:** L105 how long and what time were the spectra calculated for?

**Authors:** For the calculation of spectra, we used the whole length of records (usually about 8-9 hours). This allowed us to estimate averaged characteristics of ambient noise during data acquisition. These sentences added to the text.

**Referee:** Comparison of station spectra (fig 2) - The text implies that the spectra were calculated at different times so this doesn't seem a fair comparison and could be removed. Or if the authors want to draw some other information out they should add what times the spectra are calculated for.

**Authors:** Although the spectra were calculated for different time intervals, it is possible to compare them for obtaining qualitative information about differences in seismic noise level. One of the important features of the studied area is that only a few possible noise sources are present (Kevitsa mine with open pit, several roads, dams and river). These sources may be considered in an approximation of quasi-stationary noise sources during considered time intervals. For example, activity in Kevitsa mine is not changing during long time periods, because the same mining machinery is working, producing quasi-harmonical waves of the same amplitudes and frequencies for different times). Concerning dams, their noise can be also considered as quasi-stationary. Of course, the road

may be used by transport of different types and, as result, the noise produced by traffic may have some temporal differences. Nevertheless, most of the roads in our study area are generally characterized by low traffic (several cars per day in some cases) and as result, we can neglect this source of noise.

This explanation is added to the text.

**Referee:** L131 would be useful to define early on what is meant by heterogeneity.

**Authors:** Done

**Referee:** L137 please summarise the geology from the referenced paper and if there is evidence for mafic dykes as modelled.

**Authors:** Done

**Referee:** L162 what is the orientation of the blast from the line? Why are Rayleigh waves not produced in this case?

**Authors:** In this case, the source of the blast is located in-line with the profile of seismic sensors, because we tried to model a real situation. Kevitsa mine located about in line with Sakatti profile. Description added to the text.

**Referee:** L180 what do the authors mean by must be placed in first Fresnel zone area?

**Authors:** We mean stationary phase condition.

**Referee:** L195 is there a comparison to active source results?

**Authors:** Direct comparison is not possible, as we explained in our replies. The paragraph has been rephrased.

**Referee:** L213 is the 1D model assumed to be from directly below the node? How were the 1D models interpolated to get 2D model?

**Authors:** We used parts of profiles with lengths of 100 m and calculate average velocity models for them. Then, we applied triangular and linear interpolation to 1D models and obtained the 2D model.

**Referee:** L217 what layer thickness did Aberg etal find?

**Authors:** About 25-30 m (this has been added to the text).

**Referee:** L225 – what is the depth sensitivity of 3-7 Hz Rayleigh waves?

**Authors:** We assume that sensitivity depends on accuracy of wavelength estimation. Assuming, that the period (T) can be estimated in our case with maximum accuracy of about 0.002 s (because this is equal to the sampling period), the maximum accuracy of estimating wavelength for 3 Hz and 7 Hz would be about 1 m. Therefore, the maximum possible depths sensitivity for noise-free data is about 1 m. In reality, it is larger due to noise and error accumulation from different data processing steps (about 5-10 m).

**Referee:** L230 better to compare these models by adding a difference panel to Fig 14, difficult to compare the models as it is.

**Authors:** The difference panel has been added to the Figure 14.

**Referee:** L254 is this still part of the vibroseis signals section? Are time periods of vibroseis removed or included?

**Authors:** Yes, we used records without removing the signal of vibrosource in that case. Nevertheless, we used not the signal itself, but the scattered wavefield, produced by scattering of the signal on heterogeneities. In our case the distance between receivers was 160 m.

References:

Ikeda, T., & Tsuji, T. (2016). Surface wave attenuation in the shallow subsurface from multichannel–multishot seismic data: a new approach for detecting fractures and lithological discontinuities. Earth, Planets and Space, 68(1), 1-14.

Campillo, M., & Paul, A. (2003). Long-range correlations in the diffuse seismic coda. Science, 299(5606), 547-549.

Snieder, R., Grêt, A., Douma, H., & Scales, J. (2002). Coda wave interferometry for estimating nonlinear behavior in seismic velocity. Science, 295(5563), 2253-2255.

Snieder, R. (2006). The theory of coda wave interferometry. Pure and Applied geophysics, 163(2), 455-473.

---

## Author Response (AR3)

**Replies to Anonymous Referee #4**

**Referee:** The paper is improved compared the earlier versions. Particularly the figures are much easier to read. I am pleased to see more details of the acquisition and the comparison of the SNRS method with another method for retreiving Green's functions. The numerical modelling is better described but some questions remain, it might be worth describing in the abstract that it is not comprehensive but designed only to verify that their expected geological features could generate scattering. Also the authors should reduce claims about only the SNRS method working in this case, because they did not actually apply it to synthetic data.

**Authors:** The following sentence from the abstract explains that our modeling is not comprehensive: "To find the way to obtain EGFs, we used numerical modelling in order to investigate properties of seismic noise originating from sources with different characteristics and propagating inside synthetic heterogeneous Earth models representing real geological conditions in the XSodEx study area**".**

**Referee:** It is interested to see the comparison between the SNRS and conventional method in Fig17. The authors should describe their 'conventional' workflow used.

**Authors:** A short explanation concerning the "conventional" algorithm and references has been added to the text.

**Referee:** I still don't like the blocky velocity models. I recommend they are changed to continuous. The authors could read more into how to choose colour scales – e.g. https://www.nature.com/articles/s41467-020-19160-7

**Authors:** We are thankful to the reviewer for this reference. We modified Figures 12 and 14 and added one more figure (Fig 18 of the revised manuscript) showing the velocity models with continuous colour scale. However, we prefer also to show the models, in which we indicate the range of velocity values corresponding to different rock types. These ranges were selected based on statistically interpreted petrophysical information (Reference is in the text) and in our paper this presentation of velocity models aims to compare the velocity models to major bedrock types, not just to show the difference between sediments and bedrock.

**Referee:** In the abstract the authors claim that 'This scattered wavefield can be used to retrieve reliable Empirical Green Functions (EGFs) from short-term and non-stationary data, if using a special technique called "signal-to-noise ratio stacking" (SNRS) is applied." This should be rephrased as it is misleading, since they did not actually show that only the SNRS method will work on the synthetic data.

**Authors:** We reformulated the sentence: "This scattered wavefield can be used to retrieve reliable Empirical Green Functions (EGFs) from short-term and non-stationary data using special techniques. One of the possible solutions is application of "signal-to-noise ratio stacking" (SNRS)."

**Referee:** There are some sentences in the abstract not needed which would improve its focus.
I may have missed it – was the data acquired during the day and on weekdays?

**Authors:** The Sercel UNITE system works in such a way that the data is recorder continuously and the data corresponding to necessary time intervals are then extracted using special device called Data Harvester. Passive seismic data corresponding to nighttime and weekend were harvested from

continuous data for Sakatti profiles. The data for other lines harvested from continuous data correspond to workdays. This is described in the text.

**Referee:** L99 – what is an RAU?

**Authors:** RAU means Remote Acquisition Unit (added to the text).

**Referee:** Fig 14 – the difference colour scale is not good. Generally a difference plot should be something like white at 0 and then increase away from this point.

**Authors:** The difference colour scale was changed as proposed.

**Referee:** Fig 18 – colour scale.

**Authors:** We changed the colour scale as proposed.

**Replies to Referee #3: Rezaeifar Meysam**

**Referee:** Thanks for your response to my comments and questions. In my opinion the new version is much clear and after some minor correction of the typos (some extra spaces and dots) it would be ready.

I also have another suggestion about using MASW in the text, maybe first time you used it in the text just put the full description as "Multichannel Analysis of Surface Wave (MASW)" as there is also a code called MASW which use the same technique and that's why I was confused.

Congratulations on the work.

With kind regards,

Meysam

**Authors:** Typos have been corrected. The sentence concerning MASW has been added to the text.